# MOL-INSTRUCTIONS: A LARGE-SCALE BIOMOLECULAR INSTRUCTION DATASET FOR LLMS

**Yin Fang**♣∗, **Xiaozhuan Liang**♣∗, **Ningyu Zhang**♣†, **Kangwei Liu**♣,
**Rui Huang**♣, **Zhuo Chen**♣, **Xiaohui Fan**♣, **Huajun Chen**♣♠♡†
♣ College of Computer Science and Technology, Zhejiang University
♠ ZJU-Ant Group Joint Research Center for Knowledge Graphs, Zhejiang University
♡ ZJU-Hangzhou Global Scientific and Technological Innovation Center, Zhejiang University
{fangyin, liangxiaozhuan, kangweiliu, hrhr, zhuo.chen, fanxh}@zju.edu.cn,
{zhangningyu, huajunsir}@zju.edu.cn
**https://github.com/zjunlp/Mol-Instructions**
🤗 Dataset 🧬 Molecule model 🧫 Protein model 📄 Biotext model

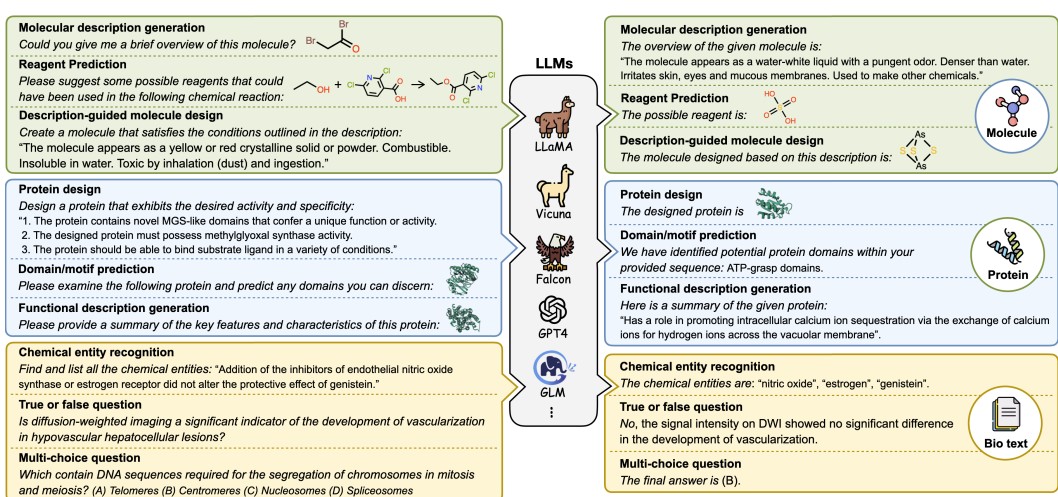

Figure 1: Empowering Large Language Models with Mol-Instructions to unlock biomolecular domain. Both molecular and protein structures are represented as sequences.

## ABSTRACT

Large Language Models (LLMs), with their remarkable task-handling capabilities and innovative outputs, have catalyzed significant advancements across a spectrum of fields. However, their proficiency within specialized domains such as biomolecular studies remains limited. To address this challenge, we introduce Mol-Instructions, a comprehensive instruction dataset designed for the biomolecular domain. Mol-Instructions encompasses three key components: molecule-oriented instructions, protein-oriented instructions, and biomolecular text instructions. Each component aims to improve the understanding and prediction capabilities of LLMs concerning biomolecular features and behaviors. Through extensive instruction tuning experiments on LLMs, we demonstrate the effectiveness of Mol-Instructions in enhancing large models' performance in the intricate realm of biomolecular studies, thus fostering progress in the biomolecular research community. Mol-Instructions is publicly available for ongoing research and will undergo regular updates to enhance its applicability.

---

∗    Equal contribution and shared co-first authorship.
†    Corresponding author.

# 1 INTRODUCTION

Large Language Models (LLMs), such as GPT-4 (OpenAI, 2023a), Chinchilla (Hoffmann et al., 2022), PaLM (Chowdhery et al., 2023), Codex (Chen et al., 2021), LLaMA (Touvron et al., 2023), FLAN (Wei et al., 2022), and GLM (Zeng et al., 2023) have revolutionized the landscape of Natural Language Processing (NLP). These models, boasting billions of parameters, are meticulously trained on extensive text corpora and excel at generating human-like text and understanding intricate contexts. To adapt LLMs for specific tasks, researchers employ instruction tuning techniques (Ouyang et al., 2022a; Sanh et al., 2022). This involves training the models with specialized instruction datasets, allowing them to acquire task-specific knowledge and patterns, thus enhancing their performance in specific domains. Numerous instruction datasets have already been developed for use in general domains. For instance, the Stanford Alpaca dataset (Taori et al., 2023) offers prompts, completions, and annotations tailored for controllable text generation. The GPT4All dataset (Anand et al., 2023) comprises a variety of formats, including code, stories, and dialogue, purposed for training and evaluating general-purpose language models. Similarly, the COIG dataset (Zhang et al., 2023a) integrates diverse corpora such as translated, exam, and human value alignment instructions, specifically oriented towards Chinese language processing.

With recent advancements, it's evident that LLMs extend beyond traditional text processing (Zhang et al., 2024; Tinn et al., 2023; Wang et al., 2023a). Their potential in biomolecular studies, covering structural biology, computational chemistry, and drug development, is particularly promising. Utilizing LLMs in this field could revolutionize our understanding and handling of biomolecular data, accelerating scientific innovations and drug discovery.

However, a major barrier to harnessing LLMs in the biomolecular domain is the lack of a dedicated dataset for this field. Despite the existence of several instruction datasets in general domains, a conspicuous gap remains when it comes to biomolecules. This gap arises from three main challenges: **First**, acquiring and annotating biomolecular data incurs significant costs, given the inherent complexity and rich depth of information contained within such data. **Second**, biomolecular computations span a broad knowledge spectrum, intertwining specialized insights from disparate areas including structural biology, computational chemistry, and drug development. **Third**, unlike the well-established frameworks in natural language processing, bioinformatics has no standardized lingua franca. Different applications often employ varied representations for biomolecules and their associated computations. This diversity amplifies the challenge of crafting a dataset that serves universally across the domain. To address this pressing need in the biomolecular domain, we introduce Mol-Instructions (CC BY-NC-SA 4.0), a dataset tailored to the unique challenges of biomolecular studies. This dataset, as delineated in Figure 1, is structured around three core components:

- **Molecule-oriented instructions:** This component delves into the realm of small molecules, emphasizing their inherent properties and behaviors. It sheds light on the fundamental challenges of diverse chemical reactions and molecular design, with 148,4K instructions across six tasks.

- **Protein-oriented instructions:** Concentrating on biosciences, it covers 505K instructions spanning five categories of tasks. These tasks aim to predict the structure, function, and activity of proteins, and facilitate protein design based on textual directives.

- **Biomolecular text instructions:** Primarily crafted for NLP tasks within the fields of bioinformatics and chemoinformatics, this portion includes six information extraction and Q&A tasks represented through 53K instructions.

Creating this instruction dataset involves gathering and collecting biomolecular data from various **licensed sources** (detailed in Appendix A.2), followed by transforming this data into easily-followable instruction formats suitable for specific tasks. Our goal is to empower LLMs with domain-specific insights, enhancing their ability to decode and predict biomolecular features. This enhancement can revolutionize the interpretation of biomolecular data, streamline drug development, and unveil new realms of biomolecular research. With Mol-Instructions, large models are endowed with the ability to comprehend biology, opening the door to new scientific discoveries. To assess the real-world effectiveness of Mol-Instructions, we conduct an extensive series of evaluations. Employing the representative LLM as the foundational model, we perform instruction tuning for each of the three main categories of instructions. The results highlight the value of Mol-Instructions, demonstrating its ability to enhance the versatility and understanding of large models in the complex domain of biomolecular studies.

Table 1: Comparison with existing datasets. We employ the following abbreviations: HG – datasets curated by human effort, SI – datasets produced via self-instruct methods, MIX – dataset composed of both human-constructed and machine-generated data, COL – dataset assembled from a variety of other datasets.

| DATASETS | # TYPE | # INSTRUCTIONS | COLLECTION | USAGE | ACCESS |
|---|---|---|---|---|---|
| *General Domain* | | | | | |
| Stanford Alpaca (Taori et al., 2023) | Text | 52,002 | SI | Instruction Tuning | Open |
| Dolly-v2 (Conover et al., 2023) | Text | 15,015 | HG | Instruction Tuning | Open |
| Baize (Xu et al., 2023) | Text | 653,699 | MIX | Instruction Tuning | Open |
| FLAN (Wei et al., 2022) | Text | 1,764,800 | COL | Instruction Tuning | Open |
| InstructGPT (Ouyang et al., 2022b) | Text | 112,801 | HG | RLHF, Instruction Tuning | Closed |
| ShareGPT (sha, 2023) | Text | 260,137 | MIX | Instruction Tuning, Chat | Closed |
| COIG (Zhang et al., 2023a) | Text | 67,798 | COL | Instruction Tuning | Open |
| UltraChat (Ding et al., 2023) | Text | 1,468,352 | MIX | Chat | Open |
| Galactica (Taylor et al., 2022) | Text, Biomolecule | 783,599 | MIX | Pre-training | Closed |
| *Specific Domain* | | | | | |
| PCdes (Zeng et al., 2022) | Text, Molecule | 15,000 | MIX | Pre-training | Closed |
| ChEBI-20 (Edwards et al., 2022) | Text, Molecule | 33,010 | COL | Pre-training | Open |
| PubChemSTM (Liu et al., 2023) | Text, Molecule | 281,000 | COL | Pre-training | Closed |
| MoMu (Su et al., 2022) | Text, Molecule | 15,000 | MIX | Pre-training | Open |
| Mol-Instructions (ours) | Text, Biomolecule | 2,043,587 | MIX | Instruction Tuning | Open |

## 2 RELATED WORK

Instruction data bears a stronger task-centric focus, typically augmenting the efficacy of LLMs utilizing a limited set of instances. Initial research efforts, such as Dolly-v2 (Conover et al., 2023) and InstructGPT (Ouyang et al., 2022b), relied heavily on manual or expert annotations to provide guidance for various NLP tasks, like closed Q&A and summarization. While human-annotated instruction data often boasts high quality, it's limited in volume, diversity, and innovation. Recognizing these limitations, there's been a shift toward semi-automated or fully automated instruction creation. For instance, Stanford Alpaca (Taori et al., 2023) employs the self-instruct approach (Wang et al., 2023b), utilizing a bootstrapping technique grounded in a set of handcrafted instructions to generate 52K diverse instructions. This innovative method has inspired numerous model-aided data collection endeavors, such as Baize (Xu et al., 2023), COIG (Zhang et al., 2023a), UltraChat (Ding et al., 2023), and ShareGPT (sha, 2023), as detailed in Table 1.

Several studies have explored the intersection of text and biomolecules (Boiko et al., 2023; Bran et al., 2023; Zeng et al., 2022; Edwards et al., 2022; Nascimento & Pimentel, 2023; Zhang et al., 2023b). For instance, Galactica (Taylor et al., 2022), a general scientific language model, incorporates a subset of instruction data related to molecules, proteins, and text during its pre-training phase. However, details of this particular dataset remain undisclosed. Other datasets, such as PCdes (Zeng et al., 2022), ChEBI-20 (Edwards et al., 2022), PubChemSTM (Liu et al., 2023), and MoMu (Su et al., 2022), pair molecules with textual descriptions. While these datasets are valuable, they are primarily geared toward training smaller models and lack an instructional format. This limitation curtails their direct utility for large language models. In contrast, Mol-Instructions covers a wider range of biomolecular tasks and a larger amount of data, achieved through a combined construction method of self-instruct, template-based conversion, and human-crafted task descriptions (as detailed in §3).

## 3 MOL-INSTRUCTIONS CONSTRUCTION

### 3.1 UNDERLYING PRINCIPLES

**Large-scale** To cater to the needs of LLMs, we've crafted Mol-Instructions to be expansive, incorporating over 2 million biomolecular instructions. This large volume provides broad and representative coverage of biomolecular sequences and structures, enabling models to grasp and navigate the complexities of biomolecules.

**Diversity** Mol-Instructions spans 17 subtasks across three types of biomolecules and includes diverse text descriptions that capture over 11 unique biomolecular properties. This extensive coverage fosters versatility in models trained with our dataset, priming them for various challenges in biomolecular research.

**Quality** We prioritize the quality of our dataset to guarantee that it serves as a reliable foundation for deriving accurate, actionable, and practical insights. Each piece of biomolecular data undergoes rigorous scrutiny to ensure its accuracy and trustworthiness.

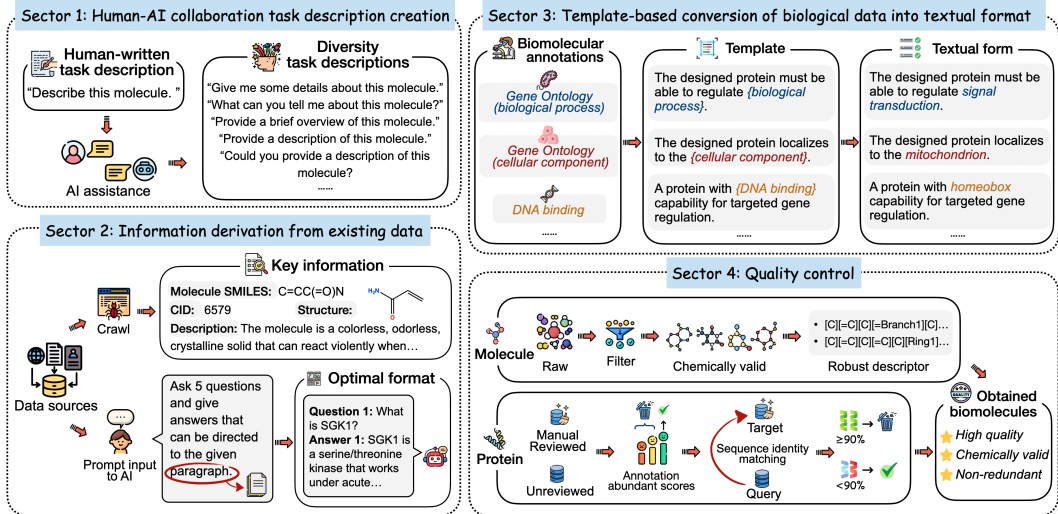

Figure 2: The overview of data construction of Mol-Instructions, which comprises four sectors: human-AI collaboration task description creation (§3.2), information derivation from existing data (§3.3), conversion of biological data into a textual format via templates (§3.4), and quality control (§3.5). For detailed procedures for each task, please refer to Appendix B.

## 3.2 HUMAN-AI COLLABORATION TASK DESCRIPTION CREATION

A standard instruction entry is typically structured with three components: an *instruction* that clarifies the task, an *input* that serves as the task's input, and an *output* that embodies the expected outcome.

In real-world scenarios, task *instructions* exhibit a broad diversity to match the dynamic and diverse nature of human inquiries and requirements. To emulate this diversity, for each task, we begin with a clear and concise human-written description, which then serves as an input to gpt-3.5-turbo (OpenAI, 2023b). Leveraging its extensive knowledge and pattern recognition skills, the LLM generates diverse task descriptions, reflecting the wide spectrum of human question-framing styles (Figure 2 Sector 1). To ensure the quality of these descriptions, each one is subjected to a thorough manual review. This collaboration of human insight and machine intelligence not only enhances the diversity and creativity of task descriptions but also bolsters the robustness and adaptability of the instructions.

## 3.3 INFORMATION DERIVATION FROM EXISTING DATA

Since biomolecular data typically involves professional wet-lab experiments and expert chemists' summaries, our data is sourced from widely-used biochemistry databases (Kim et al., 2021; Wei et al., 2010; Lu & Zhang, 2022; Wu et al., 2018; Ashburner et al., 2000; Consortium, 2023; Krallinger et al., 2015; 2017; Li et al., 2016; Pal et al., 2022; Hendrycks et al., 2021). Based on these data sources, we can obtain the desired instruction data with appropriate processing.

Some datasets undergo manual processing, with labels being directly appended to specific biomolecular data. This includes datasets with clearly delineated inputs and anticipated outcomes for various prediction endeavors, as well as datasets with predetermined questions and corresponding answers for Q&A tasks. Processing such data is fairly direct: the labeled information is typically mapped to the respective *input* and *output* fields for each instruction entry.

Another class of data sources comes without explicit manual labels and requires techniques such as data mining and AI-assisted generation for extracting and selecting the ideal data, as depicted in Figure 2 Sector 2. With data mining, we strive for data comprehensiveness and adequacy by extracting additional relevant information from professional chemical research databases like PubChem (Kim et al., 2021). Specifically, we crawl valid molecule description texts and their corresponding PubChem Chemical Identifiers (CIDs), followed by retrieving the respective molecular descriptors. To focus the model's attention on description semantics, we replace the molecular nomenclature with the term "the molecule". On the AI-assisted generation front, we use scientific abstracts from PubMed (White, 2020) to generate Open Question instructions. Here, we task gpt-3.5-turbo to formulate questions and their corresponding answers based on the abstracts, structured as Q&A pairs. This methodology

efficiently produces a varied collection of Q&A instructions, simplifying the construction of this particular task. For a more task-level detailed explanation, please refer to Appendix B.

### 3.4 TEMPLATE-BASED CONVERSION OF BIOLOGICAL DATA INTO TEXTUAL FORMAT

Certainly, not all data can be seamlessly transformed into ideal instruction sets. For some novel tasks, finding directly applicable data can be particularly challenging. This is especially true for protein design tasks, where previous studies often conditioned designs on broad categories (Madani et al., 2023) or fixed backbones (Dauparas et al., 2022), largely neglecting the bespoke functional and structural attributes (e.g., helical secondary structure, DNA binding domain, or transcription regulator activity). Conversely, we aim to devise instructions in textual form for *de novo* protein design tailored to user intent, enabling the design of composite properties of interest. Given the scarcity of directly applicable data for such tasks, we curate annotations from specific select within UniProtKB (Consortium, 2023). These annotations encapsulate the protein properties frequently explored or sought after by researchers in the field (detailed examples are in Appendix Table 6).

To effectively convert these structured annotations into textual format, we formulate a series of templates, illustrated in Figure 2 Sector 3 and Appendix Table 7. Each resulting textual annotation establishes criteria for protein design. By aggregating these functional descriptions and properties, we craft precise protein design instructions, ensuring that the synthesized protein meets the specified criteria. In practice, considering the model's input length limitations and training efficiency, we randomly select a subset of these conditions as design objectives.

### 3.5 QUALITY CONTROL

As the field stands, LLMs have not yet fully grasped the intricacies of biomolecular languages, falling short of their proficiency with human languages. To accelerate the model's capability to generate accurate biomolecules, we implement stringent quality assurance measures for our biomolecular data, detailed in Figure 2 Sector 4.

For small molecules, our process begins by eliminating chemically invalid SMILES strings from the initial dataset. Although external constraints (Landrum et al., 2013) can validate generated molecules, using a dependable molecular descriptor within molecular instructions is more effective. While SMILES (Weininger, 1988) strings remain a popular choice for molecular descriptors, models leveraging them often output strings that are either syntactically flawed or chemically inconsistent. To circumvent these issues, we opt for SELFIES (Krenn et al., 2022) as our molecular descriptor. With its tight rule set, SELFIES ensures the generation of valid molecules by allowing combinations of any symbols, eliminating common pitfalls associated with SMILES strings, such as producing illogical symbols or mismatched parentheses.

Alongside, we prioritize the integrity of our protein data by primarily sourcing entries from UniProtKB/Swiss-Prot (Consortium, 2023), a curated and manually annotated protein sequence database. To bolster the volume and variety of our data, we supplement with high-scoring annotations from UniProtKB/TrEMBL. Recognizing the risk of bias from redundant protein sequences, such as homologous sequences or closely-related protein variants within the Swiss-Prot and TrEMBL databases, we deploy a meticulous filtration process. Leveraging the MMseqs (Steinegger & Söding, 2017) tool, we cluster protein sequences at a 90% similarity threshold, selecting functionally rich entries within each cluster as representatives and excluding the rest. This rigorous approach ensures our dataset comprises diverse, high-quality, and function-centric protein instructions devoid of redundancies.

## 4 A CLOSER LOOK AT MOL-INSTRUCTIONS

### 4.1 CATEGORIZATION AND POTENTIAL APPLICATIONS OF INSTRUCTION TASKS

As illustrated in Figure 3, Mol-Instructions is anchored around three core domains: molecule-oriented, protein-oriented, and biomolecular texts. Each category's instruction task covers the essential challenges within its domain, aspiring to drive forward the biomolecular field via the

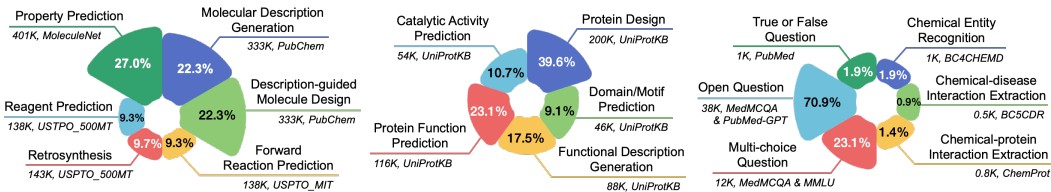

(a) Molecule-oriented Instructions     (b) Protein-oriented Instructions     (c) Biomolecular Text Instructions

Figure 3: The compositional structure of Mol-Instructions. Mol-Instructions primarily encompasses tasks across three major categories: (a), (b) and (c). The task names are indicated above the horizontal lines, the sources of the original data and the sizes of the constructed instruction datasets are labeled below the horizontal lines, and the percentages on the pie charts represent the proportion of data within each major category.

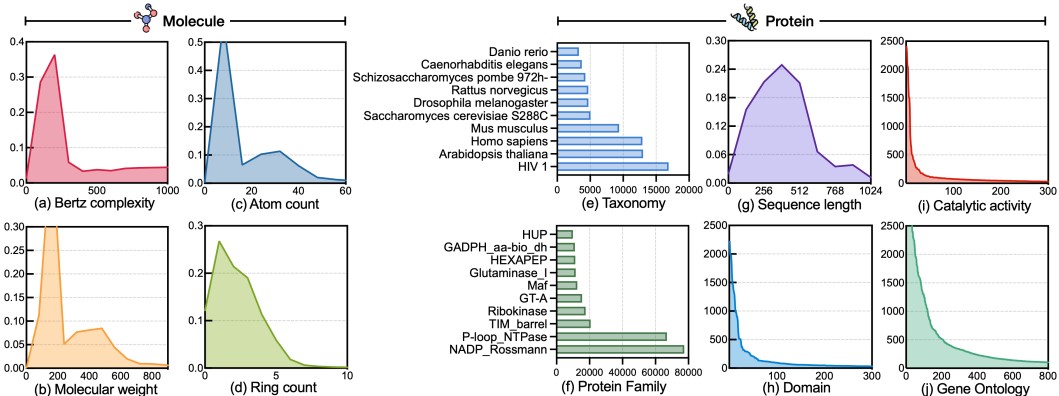

Figure 4: Multidimensional analysis of biomolecular sequences. The left side illustrates the diversity of molecules within Mol-Instructions, encompassing molecules of varying complexity and distinct structures. On the right side, the analysis delves into the diversity of protein sequences within Mol-Instructions, considering different aspects such as sequence length, domain, and activity.

training and utilization of LLMs. A comprehensive exposition on task definitions and instruction construction is provided in Appendix B.

**Molecule-oriented instructions** center on small molecules, delving into their natural properties and behaviors. It underscores the foundational challenges of various chemical reactions and molecular design, encompassing six tasks illustrated in Figure 3 (a). The objective is to understand and predict the chemical properties of molecules, improve molecule design, and increase the accuracy and speed of chemical reactions. In the areas of chemical and pharmaceutical design, the predictions from these tasks could expedite drug innovation and diminish developmental expenses.

**Protein-oriented instructions** are rooted in bioscience, primarily addressing issues pertinent to protein design and functionality. This segment spans five distinct categories, as detailed in Figure 3 (b). The endeavors here are directed towards predicting protein domains, functions, and activities, and facilitating protein design via textual instructions. Understanding the fundamentals of protein function and folding is crucial for realms such as disease diagnosis, treatment, and novel drug discovery.

**Biomolecular text instructions** concentrate primarily on NLP tasks related to bioinformatics and chemoinformatics. This category contains six information extraction and Q&A tasks as shown in Figure 3 (c). The goals here are to interpret and extract pivotal information from biomedical literature, aiding researchers in swiftly accumulating insights and propelling their investigative pursuits.

## 4.2 Diversity and Complexity of Biomolecular Traits

Figure 4 provides a comprehensive overview of the biomolecule distribution across different dimensions. For clarity in visualization, we've truncated the long-tail segments of the data, preserving its core essence. An exhaustive analysis is available in Appendix C.

Figure 4 (a-d) showcases the diverse characteristics of **molecules**. Bertz complexity serves as a crucial metric for assessing molecular complexity. Molecular weight, indicative of a molecule's scale and complexity, is instrumental in many chemical reactions. The atom count provides insight into a molecule's dimension and complexity, influencing its stability and reactivity. The ring count offers a

Table 2: The predominant biomolecular characteristics encapsulated within the textual descriptions.

| | Features | Example |
|---|---|---|
| **Molecule** | Chemical properties | It combines with metals to make fluorides such as sodium fluoride and calcium fluoride. |
| | Physical properties | The molecule is a colorless, flammable gas that has a distinct, pungent smell. |
| | Applications | Used as a flavoring, solvent, and polymerization catalyst. |
| | Environment | The molecule is a metal that occurs naturally throughout the environment, in rocks, soil, water, and air. |
| | Safety | Lethal by inhalation and highly toxic or lethal by skin absorption. |
| | Formation | It is formed in foods that are rich in carbohydrates when they are fried, grilled, or baked. |
| **Protein** | Function | The designed protein must be able to regulate signal transduction. |
| | Subcellular location | The designed protein localizes to the mitochondrion. |
| | Structure | The target protein must exhibit Helix as its primary conformation. |
| | Family & Domain | The designed protein should contain PWWP domain that is essential for its function. |
| | PTM / Processing | Incorporate a signal peptide in the protein design. |

lens into structural complexity and potential stability, with implications for chemical reactivity and probable biological activity.

In Figure 4 (e-j), we delve into the attributes of **proteins** within Mol-Instructions. Figure 4 (e-g) highlights the varied distribution of protein sequence lengths. Categorized according to the NCBI Taxonomy, these proteins encompass an expansive range of species and experimental strains, encapsulating 13,563 protein families and 643 superfamilies. Figures 4 (h-j) spotlight functional facets, such as domain, gene ontology, and catalytic activity annotations. The data exhibits a pronounced long-tail distribution, underscoring challenges in inferring functions for proteins, especially those with infrequent functions.

### 4.3 EXTENSIVE COVERAGE OF BIOMOLECULAR DESCRIPTIONS

In the burgeoning field of text-driven biomolecular design, our emphasis lies on the depth and diversity of biomolecular description texts.

As shown in Table 2, **molecular** text descriptions provide a comprehensive, multi-dimensional, and in-depth view of molecular information. Regarding information depth and breadth, the listed molecular properties offer a wide perspective, ranging from basic chemical attributes to specific application contexts. Such comprehensive coverage allows text descriptions to depict molecules in a varied and layered fashion. By understanding the expressed chemical and physical characteristics, one can grasp the essential features and reactive tendencies of molecules. Further, insights into their applications, environmental prevalence, and safety aspects offer a holistic understanding of their significance and associated safety considerations.

As detailed in Section §3.4, we convert biological insights and functional annotations of naturally occurring proteins into text-based design specifications. Table 2 highlights the diverse features of these **proteins**, considering five related aspects that cover their behavior in protein folding, maturation, processing, and their contribution to life processes. Unlike traditional *de novo* protein design that emphasizes protein generation grounded in physical principles, Mol-Instructions targets the creation of proteins with multifaceted desired traits. This challenges the models' capability to discern the intricate sampling space defined by the convergence of multiple attributes.

## 5 EXPLORING THE POTENTIAL OF MOL-INSTRUCTIONS

### 5.1 INSIGHTS FROM PERFORMANCE ANALYSIS

To investigate whether Mol-Instructions can enhance LLM's understanding of biomolecules, we perform instruction tuning on the three main domains of Mol-instructions, using LLama-7B (Touvron et al., 2023) as the foundation model. We additionally employ Alpaca-LoRA (Tloen, 2023), Baize-7B (Xu et al., 2023), ChatGLM-6B (Zeng et al., 2023), Vicuna (Chiang et al., 2023), Galactica (Taylor et al., 2022), Text+Chem T5 (Christofidellis et al., 2023), MolT5 (Edwards et al., 2022), and PMC-LLaMA-13B (Wu et al., 2023) as baselines. Our dataset is partitioned into training, validation, and testing subsets. The training and validation sets are used for instruction tuning, while the test set assesses model performance. For detailed training procedures, evaluation metrics, and case studies, please refer to Appendix D, E, and F.

Table 3: Results of molecular property prediction tasks.

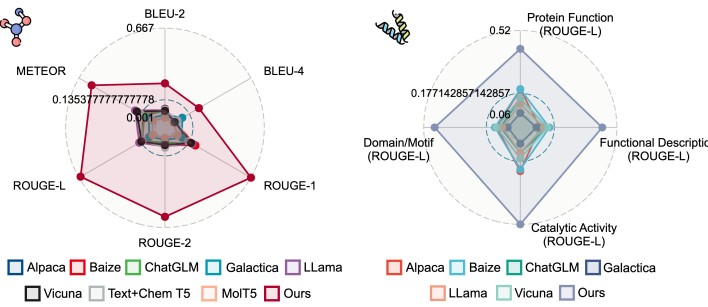

| MODEL | MAE↓ |
|---|---|
| *Property Prediction* | |
| ALPACA | 322.109 |
| BAIZE | 261.343 |
| CHATGLM | - |
| LLAMA | 5.553 |
| VICUNA | 860.051 |
| GALACTICA | 0.568 |
| OURS | ↑0.555 **0.013** |

Figure 5: The performance comparison on molecule and protein understanding tasks: molecule description generation (left), protein function, functional description, catalytic activity, and domain/motif prediction (right).

Table 4: Results of molecular generation tasks. These tasks encompass description-guided molecule design, reagent prediction, forward reaction prediction, and retrosynthesis.

| MODEL | EXACT↑ | BLEU↑ | LEVENSHTEIN↓ | RDK FTS↑ | MACCS FTS↑ | MORGAN FTS↑ | VALIDITY↑ |
|---|---|---|---|---|---|---|---|
| *Description-guided Molecule Design* | | | | | | | |
| ALPACA | 0.000 | 0.004 | 51.088 | 0.006 | 0.029 | 0.000 | 0.002 |
| BAIZE | 0.000 | 0.006 | 53.796 | 0.000 | 0.000 | 0.000 | 0.002 |
| CHATGLM | 0.000 | 0.004 | 53.157 | 0.005 | 0.000 | 0.000 | 0.005 |
| LLAMA | 0.000 | 0.003 | 59.864 | 0.005 | 0.000 | 0.000 | 0.003 |
| VICUNA | 0.000 | 0.006 | 60.356 | 0.006 | 0.001 | 0.000 | 0.001 |
| GALACTICA | 0.000 | 0.192 | 44.152 | 0.135 | 0.248 | 0.088 | 0.992 |
| TEXT+CHEM T5 | 0.097 | 0.508 | 41.819 | 0.352 | 0.474 | 0.353 | 0.721 |
| MOLT5 | 0.112 | 0.546 | 38.276 | 0.400 | 0.538 | 0.295 | 0.773 |
| **OURS** | 0.002 | 0.345 | 41.367 | 0.231 | 0.412 | 0.147 | 1.000 |
| *Reagent Prediction* | | | | | | | |
| ALPACA | 0.000 | 0.026 | 29.037 | 0.029 | 0.016 | 0.001 | 0.186 |
| BAIZE | 0.000 | 0.051 | 30.628 | 0.022 | 0.018 | 0.004 | 0.099 |
| CHATGLM | 0.000 | 0.019 | 29.169 | 0.017 | 0.006 | 0.002 | 0.074 |
| LLAMA | 0.000 | 0.003 | 28.040 | 0.037 | 0.001 | 0.001 | 0.001 |
| VICUNA | 0.000 | 0.010 | 27.948 | 0.038 | 0.002 | 0.001 | 0.007 |
| GALACTICA | 0.000 | 0.141 | 30.760 | 0.036 | 0.127 | 0.051 | 0.995 |
| TEXT+CHEM T5 | 0.000 | 0.225 | 49.323 | 0.039 | 0.186 | 0.052 | 0.313 |
| **OURS** | 0.044 | 0.224 | 23.167 | 0.237 | 0.364 | 0.213 | 1.000 |
| *Forward Reaction Prediction* | | | | | | | |
| ALPACA | 0.000 | 0.065 | 41.989 | 0.004 | 0.024 | 0.008 | 0.138 |
| BAIZE | 0.000 | 0.044 | 41.500 | 0.004 | 0.025 | 0.009 | 0.097 |
| CHATGLM | 0.000 | 0.183 | 40.008 | 0.050 | 0.100 | 0.044 | 0.108 |
| LLAMA | 0.000 | 0.020 | 42.002 | 0.001 | 0.002 | 0.001 | 0.039 |
| VICUNA | 0.000 | 0.057 | 41.690 | 0.007 | 0.016 | 0.006 | 0.059 |
| GALACTICA | 0.000 | 0.468 | 35.021 | 0.156 | 0.257 | 0.097 | 0.946 |
| TEXT+CHEM T5 | 0.239 | 0.782 | 20.413 | 0.705 | 0.789 | 0.652 | 0.762 |
| **OURS** | 0.045 | 0.654 | 27.262 | 0.313 | 0.509 | 0.262 | 1.000 |
| *Retrosynthesis* | | | | | | | |
| ALPACA | 0.000 | 0.063 | 46.915 | 0.005 | 0.023 | 0.007 | 0.160 |
| BAIZE | 0.000 | 0.095 | 44.714 | 0.025 | 0.050 | 0.023 | 0.112 |
| CHATGLM | 0.000 | 0.117 | 48.365 | 0.056 | 0.075 | 0.043 | 0.046 |
| LLAMA | 0.000 | 0.036 | 46.844 | 0.018 | 0.029 | 0.017 | 0.010 |
| VICUNA | 0.000 | 0.057 | 46.877 | 0.025 | 0.030 | 0.021 | 0.017 |
| GALACTICA | 0.000 | 0.452 | 34.940 | 0.167 | 0.274 | 0.134 | 0.986 |
| TEXT+CHEM T5 | 0.141 | 0.765 | 24.043 | 0.685 | 0.765 | 0.585 | 0.698 |
| **OURS** | 0.009 | 0.705 | 31.227 | 0.283 | 0.487 | 0.230 | 1.000 |

Assessing the accuracy of LLM-generated results in the life sciences field is inherently complex. Subjecting every output to expert review or wet-lab validation would be both time-consuming and impractical. Moreover, given that LLMs can produce varied outputs for the same input, ensuring uniform accuracy across all potential outcomes is a formidable challenge. While we utilize metrics commonly accepted in broader domains to evaluate output quality, these metrics capture only a slice of the overall picture. Quantitative experiments may shed light on performance on certain tasks, but they fall short of comprehensively addressing the evaluation challenges.

As shown in Figure 5, Mol-Instructions enhances the molecular understanding capabilities of LLMs, demonstrating notable improvements in every metric compared to baseline models, including even domain-specific smaller models. Notably, LLMs exhibit an impressive aptitude for predicting molecular properties, as detailed in Figure 3. In this specific task, Alpaca generated answers for a mere 2.62% of the samples, Baize for 0.62%, LLama for 54.5%, Vicuna for 0.14%, Galactica for 74%, whereas ChatGLM refrained from responding. Regarding molecular generation tasks, as depicted in Table 4, LLMs exhibit the capability to generate valid molecules, and compared to the baseline, these generated molecules exhibit a higher degree of similarity to reference molecules. This demonstrates

Figure 6: Results for bioinformatic NLP tasks. (C)ER denotes chemical entity recognition, (C-D)IE signifies chemical-disease interaction extraction, and (C-P)IE stands for chemical-protein interaction extraction.

that Mol-Instructions equips LLMs with new insights into molecular generation, chemical reaction prediction, and the synthesis of molecules based on specific instructions. However, the molecule generation capabilities of LLMs still exhibit a discernible gap when compared to specialized smaller models, mainly because LLMs are designed to handle a wider range of tasks at the expense of the specialized performance seen in more focused models.

As demonstrated in Figure 5, for various protein understanding tasks, the LLM tuned with Mol-Instructions can analyze proteins in accordance with specific requirements and exhibits a noteworthy capability in identifying fundamental protein characteristics. For the protein design task, the determination of target functional features for the generated sequences is challenging to validate experimentally. Therefore, as shown in Appendix F, we employ a straightforward approach of aligning the *de novo* protein sequence to the UniProtKB using BLAST (Camacho et al., 2009). We observe a significant sequence identity between the generated sequence and multiple proteins in the corresponding functional regions. Specifically, we identify the protein (UniProt Accession: *A0A518LQL6*) as the target protein to perform alignment, as it exhibits the highest sequence identity (40.9%, with a p-value of $7.7e-30$) with the generated sequence. Notably, the *A0A518LQL6* protein and the generated protein share similar sequence signatures in the (6S)-NADPHX and metal ion binding regions (residues 1-200). This finding suggests that the generated protein fulfills the design requirements and has the potential for NADPHX epimerase activity.

Figure 6 shows the model's proficiency in comprehending biomolecular text instructions. By infusing the LLM with substantial knowledge from the biomolecular domain, the model demonstrates superior performance compared to the baseline in all NLP tasks. This further underscores the transformative potential of Mol-Instructions in bridging the gap between LLMs and intricate biomolecular studies.

## 5.2 HARNESSING THE POWER OF MOL-INSTRUCTIONS

To optimize researchers' utilization of Mol-Instructions, we suggest three pivotal directions to enhance general model exploration and drive progress in biomolecular understanding and drug design: First, use Mol-Instructions to **assess cross-modal comprehension** in general models, transitioning from human to life languages. These models should interpret user intentions and decode the biomolecular language, challenging their reasoning capabilities. Second, our work lays the groundwork for deeper **biomolecular design exploration**. Mol-Instructions cover a wide range of design criteria, by providing data related to biomolecular property prediction tasks, bolsters model understanding of biomolecules. Third, employ Mol-Instructions as **essential data for tool learning in addressing complex biological issues**. Though research highlights the advantages of specialized tools with foundational models (Qin et al., 2023), a gap exists in the availability of textual instructions, which our dataset addresses.

## 6 CONCLUSION AND FUTURE WORK

In this work, we introduce Mol-Instructions, a comprehensive instruction dataset specifically curated for biomolecular studies, bridging the gap in current resources and advancing LLM training in this specialized sphere. Looking ahead, we are committed to the ongoing enrichment and refinement of Mol-Instructions. We will incorporate a wider array of task types, instruction entries, and modalities, in line with the latest advances in chemical research and improvements in AI technology to meet broader chemical research needs and higher-level LLM training requirements. Moreover, given the distinct representation spaces of text and biomolecules, coupled with the limitations imposed by LoRA's training strategy, current LLMs have yet to master the biomolecular language as proficiently as they do human language. Exploring methods to expand the vocabulary, or incorporating bio language as a modality via biomolecular encoders (Fang et al., 2024; Rives et al., 2021; Lin et al., 2022; Cao et al., 2023; Pei et al., 2024), could be pivotal in honing the model's understanding and performance in biomolecular tasks.

## ACKNOWLEDGMENTS

We would like to express gratitude to the anonymous reviewers for kind comments. This work was supported by the National Natural Science Foundation of China (No. 62206246), the Fundamental Research Funds for the Central Universities (226-2023-00138), Zhejiang Provincial Natural Science Foundation of China (No. LGG22F030011), Ningbo Natural Science Foundation (2021J190), CAAI-Huawei MindSpore Open Fund, Yongjiang Talent Introduction Programme (2021A-156-G), CCF-Baidu Open Fund, and Information Technology Center and State Key Lab of CAD&CG, Zhejiang University.

## REPRODUCIBILITY STATEMENT

All data, code, and model weights can be found on GitHub [1] and Hugging Face [2,3,4,5]. For a detailed description of the dataset construction process, please refer to Appendix B. For specific experimental settings, please see Appendix D and E.

## ETHICS STATEMENT

This study was carried out in strict accordance with ethical guidelines and best practices in research. The biomolecular data utilized were sourced from publicly available datasets, and no proprietary or confidential data were used. Furthermore, we have obtained the necessary permissions and licenses for all third-party content incorporated in our dataset, as detailed in Appendix A.2.

Rigorous quality control measures and security checks have been implemented to preclude the presence of any harmful or malicious content in our dataset. However, we recognize the profound implications and potential risks associated with the integration of LLMs and biomolecular knowledge. While our primary intent is to advance scientific understanding and contribute positively to society, we are acutely aware that these tools, in the wrong hands, could be misused. There exists the potential for malicious actors to harness the combined capabilities of LLMs and biomolecular data to generate harmful substances, such as biochemical weapons or illicit drugs.

We strongly urge all users to adhere to the highest ethical standards when using our dataset, ensuring fairness, transparency, and responsibility in their research. Any usage of the dataset that may lead to harm or pose a detriment to society is strictly forbidden.

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

## A  ACCESSING AND UTILIZING MOL-INSTRUCTIONS

### A.1  HOSTING AND ACCESS DETAILS

Our dataset and associated models are securely hosted on GitHub and Hugging Face, which are well-recognized platforms for managing open-source projects. They provide vast accessibility and efficient management of extensive data and code repositories, guaranteeing unobstructed access to all potential users. To enable efficient usage of our resources, we provide comprehensive guidelines and instructions on the repository. These cover how to explore the dataset, understand its structure, and utilize the models.

### A.2  DATA SOURCES AND LICENSE

As shown in Table 5, we detail the sources and data rights for all data components used in constructing the Mol-Instructionsdataset, including both biomolecules and textual descriptions. All data sources were rigorously scrutinized to ensure their licenses permit our kind of research and subsequent usage. Throughout the article, every mention or use of these data sources is properly and accurately cited.

### A.3  USAGE GUIDELINES AND OBLIGATIONS

We assert that all data included in this work comply with the stipulations of the CC BY 4.0 License. We acknowledge our responsibility as licensors to facilitate open and fair use of the dataset while recognizing the creators' contributions. We accept all obligations related to enforcing this license and pledge our commitment to assist all potential users in understanding their rights and responsibilities under this license.

We assure that our dataset does not contain any personally identifiable or privacy-sensitive information. We have enforced stringent quality control procedures and security checks to prevent the inclusion of harmful or malicious content. However, it is important to note that while we have taken considerable measures to ensure the dataset's safety and privacy, we do not bear responsibility for any issues that may arise from its use.

We emphatically urge all users to adhere to the highest ethical standards when using our dataset, including maintaining fairness, transparency, and responsibility in their research. Any usage of the dataset that may lead to harm or pose a detriment to society is strictly forbidden.

In terms of dataset maintenance, we pledge our commitment to provide necessary upkeep. This will ensure the continued relevance and usability of the dataset in light of evolving research landscapes. This commitment encompasses regular updates, error checks, and amendments in accordance with field advancements and user feedback.

## B  TASK DEFINITION AND DATA CONSTRUCTION

### B.1  MOLECULE-ORIENTED INSTRUCTIONS

**Molecular description generation**   Molecular description generation entails the creation of a detailed textual depiction illuminating the structure, properties, biological activity, and applications of a molecule based on its molecular descriptors. It furnishes chemists and biologists with a swift conduit to essential molecular information, thus efficiently guiding their research and experiments.

To gather ample and authoritative molecular text annotation data, we chose PubChem (Kim et al., 2021) as the data source. PubChem, a freely accessible database managed by the National Center for Biotechnology Information (NCBI), is a valuable resource for chemical research. Many compounds in PubChem have text descriptions that come from direct submissions from various research institutions, and automated data mining and information extraction from scientific literature and patents.

Firstly, we commence with PubChem's Power User Gateway, which provides abstracts of PubChem compound records in XML format, facilitating the efficient retrieval and processing of chemical information. We crawl all valid molecular description texts along with the unique PubChem Chemical

Table 5: Data resources and licenses involved in our paper.

| Data Sources | License URL | License Note |
|---|---|---|
| PubChem | https://www.nlm.nih.gov/web_policies.html | Works produced by the U.S. government are not subject to copyright protection in the United States. Any such works found on National Library of Medicine (NLM) Web sites may be freely used or reproduced without permission in the U.S. |
| USPTO | https://www.uspto.gov/learning-and-resources/open-data-and-mobility | It can be freely used, reused, and redistributed by anyone. |
| UniProtKB | https://www.uniprot.org/help/license | You are free to: Share — copy and redistribute the material in any medium or format. Adapt — remix, transform, and build upon the material for any purpose, even commercially. |
| BC4CHEMD, ChemProt, BC5CDR | https://biocreative.bioinformatics.udel.edu/tasks/biocreative-v/track-3-cdr/ | The task data is freely available now for the research community. |
| MoleculeNet, MMLU, PubMedQA, MedMCQA | https://opensource.org/license/mit/ | Permission is hereby granted, free of charge, to any person obtaining a copy of this software and associated documentation files (the "Software"), to deal in the Software without restriction, including without limitation the rights to use, copy, modify, merge, publish, distribute, sublicense, and/or sell copies of the Software, and to permit persons to whom the Software is furnished to do so. |
| Agency for Toxic Substances and Disease Registry (ATSDR) | https://www.cdc.gov/Other/disclaimer.html | This site uses the AddThis service to allow visitors to bookmark and share website content on a variety of social media sites. Visitors who use the AddThis service to share content do not need to register or provide any personal information. |
| FDA Pharm Classes | https://www.fda.gov/about-fda/about-website/website-policies | Unless otherwise noted, the contents of the FDA website (www.fda.gov), both text and graphics, are not copyrighted. They are in the public domain and may be republished, reprinted and otherwise used freely by anyone without the need to obtain permission from FDA. Credit to the U.S. Food and Drug Administration as the source is appreciated but not required. |
| LiverTox | https://www.nlm.nih.gov/copyright.html | Works produced by the U.S. government are not subject to copyright protection in the United States. Any such works found on National Library of Medicine (NLM) Web sites may be freely used or reproduced without permission in the U.S. |
| Drug Database, Clinicalinfo.hiv.gov | https://www.hiv.gov/about-us/mission-and-team | Unless otherwise noted, material presented on the HIV.gov website is considered Federal government information and is in the public domain. That means this information may be freely copied and distributed. |
| Drug Bank | https://creativecommons.org/licenses/by-nc/4.0/legalcode | Subject to the terms and conditions of this Public License, the Licensor hereby grants You a worldwide, royalty-free, non-sublicensable, non-exclusive, irrevocable license to exercise the Licensed Rights in the Licensed Material to: reproduce and Share the Licensed Material, in whole or in part, for NonCommercial purposes only; and produce, reproduce, and Share Adapted Material for NonCommercial purposes only. |
| ChEBI | https://creativecommons.org/licenses/by/4.0/ | You are free to: Share — copy and redistribute the material in any medium or format. Adapt — remix, transform, and build upon the material for any purpose, even commercially. |
| Yeast Metabolome Database (YMDB) | http://www.ymdb.ca/downloads | YMDB is offered to the public as a freely available resource. |
| LOTUS - the natural products occurrence database | https://lotus.nprod.net/ | LOTUS is one of the biggest and best annotated resources for natural products occurrences available free of charge and without any restriction. |
| GlyCosmos Glycoscience Portal | https://glycosmos.org/license | You are free to: Share — copy and redistribute the material in any medium or format. Adapt — remix, transform, and build upon the material for any purpose, even commercially. |
| CAMEO Chemicals | https://cameochemicals.noaa.gov/help/reference/terms_and_conditions.htm?d_f=false | CAMEO Chemicals and all other CAMEO products are available at no charge to those organizations and individuals (recipients) responsible for the safe handling of chemicals. |
| E. coli Metabolome Database (ECMDB) | https://ecmdb.ca/citations | ECMDB is offered to the public as a freely available resource. |

Identifier (CID) corresponding to the molecule, and employ the CID to retrieve the corresponding SMILES representation from all compounds documented in PubChem. Secondly, given that the SMILES provided by PubChem may not always be accurate, we filter out SMILES that contain

syntactic errors or violate fundamental chemical principles, and convert all valid SMILES into SELFIES format. Note that we carry out this step in every task, and it will not be reiterated in the subsequent sections. Thirdly, to compel the model to focus on the semantics of the description, we follow Edwards et al. (2022) to replace the molecule's name in every description with "the molecule is...". Finally, we call the gpt-3.5-turbo to generate a diverse set of task descriptions via prompts, which we then randomly assign to each SELFIES-description pair. We have compiled a total of 331,261 instructions.

**Description-guided molecule generation**   The significance of description-based molecule generation lies in its potential to streamline the process of molecular design by enabling the production of molecules that directly meet the criteria outlined in a given description. This facilitates a more targeted approach in the creation and optimization of novel molecules, with applications in diverse fields such as drug discovery and materials science.

In this phase, we repurpose the SELFIES-description pairs amassed from PubChem in the preceding task. Unlike before, in this instance, the instruction entries have the molecular description as the *input* and the SELFIES representation of the molecule as the *output*. We subsequently fabricate a series of task descriptions for description-guided molecule generation serving as the *instruction*. This results in a compilation of 331,261 instruction data entries.

**Forward reaction prediction**   Forward reaction prediction pertains to the anticipatory determination of the probable product(s) of a chemical reaction, given specific reactants and reagents. This facilitates the optimization of research and development methodologies, curbs the extent of experimental speculation, and endorses greener chemistry practices by mitigating waste production.

To collect high-quality chemical reaction data, we focus on the USPTO dataset (Wei et al., 2010). This dataset encompasses a multitude of organic chemical reactions in SMILES format, extracted from American patents and patent applications. The general format is "reactant ≻ reagent ≻ product".

Typically, reagents are defined as chemical substances that do not appear in the main product. However, in an effort to simulate the real-world scenario of non-chemically specialized individuals, we undertake a more challenging task, one that does not separate reagents and reactants but requires the model to identify them independently. After preprocessing, we secure 138,768 standard instruction data entries. In each of these instruction entries, the *input* signifies the reactants and reagents in the reaction, with individual chemicals separated by a period ('.'). Conversely, the *output* denotes the product of this reaction.

**Retrosynthesis**   Retrosynthetic analysis is a pivotal synthetic methodology in organic chemistry that employs a reverse-engineering approach, initiating from the target compound and retroactively tracing potential synthesis routes and precursor molecules. This technique proves instrumental in sculpting efficient synthetic strategies for intricate molecules, thus catalyzing the evolution and progression of novel pharmaceuticals and materials.

In this task, we concentrate exclusively on single-step retrosynthesis. Our data collection comes from the manually processed USPTO_500MT dataset (Lu & Zhang, 2022), which itself is meticulously refined from the USPTO database (Wei et al., 2010). Through preprocessing, we have obtained 143,536 standard instruction entries. In each entry, the *input* is the product, and the *output* is the reactants, with each reactant separated by a period ('.').

**Reagent prediction**   Reagent prediction endeavors to ascertain the suitable catalysts, solvents, or ancillary substances required for a specific chemical reaction. This endeavor facilitates chemists in uncovering novel reaction types and mechanisms, identifying more optimal or eco-friendly reaction conditions, and ultimately streamlining the comprehensive chemical process to attain maximal cost-effectiveness and environmental stewardship.

Like retrosynthesis, the data for this task also originates from the USPTO_500K dataset (Lu & Zhang, 2022). After processing, we have approximately 138,768 entries. In each instruction entry, the *input* is a chemical reaction from reactants to product, formatted as "reactant ≻≻ product". The expected *output* is the potential reagents that facilitate this reaction.

**Property prediction**    Property prediction involves forecasting or estimating a molecule's inherent physical and chemical properties based on information derived from its structural characteristics. It facilitates high-throughput evaluation of an extensive array of molecular properties, enabling the virtual screening of compounds. Additionally, it provides the means to predict the unknown attributes of new molecules, thereby bolstering research efficiency and reducing development times.

In this task, we primarily focus on the quantum mechanics properties of molecules, with data drawn from the QM9 dataset of MoleculeNet (Wu et al., 2018). The dataset is replete with multiple property data associated with each molecule, leading to the creation of a distinctive instruction for every property linked to its corresponding molecule. For example, given molecule $\mathcal{A}$ and its properties $\mathcal{P}_1$, $\mathcal{P}_2$, and $\mathcal{P}_3$, we generate three distinct instructions, with each entry featuring molecule $\mathcal{A}$ as the *input* and the respective property ($\mathcal{P}_1$, $\mathcal{P}_2$, or $\mathcal{P}_3$) as the *output*. Correspondingly, individual instructions mirror the property they represent. For a comprehensive understanding, please refer to Figure 8 (a). Following this procedure, we have derived 401,229 instruction entries after preprocessing.

Certainly, we acknowledge that the scope of properties presently collected might be somewhat narrow, not fully representing the comprehensive landscape of the molecular domain. Efforts will be made to continually optimize and enrich our instruction dataset. The initial intent of creating such a type of instruction is to investigate whether LLMs can yield pertinent *output* given identical *input*, but with varying *instruction* demands.

### B.2    PROTEIN-ORIENTED INSTRUCTIONS

UniProtKB, a universal protein knowledgebase, is used as our data source. Concurrently, to safeguard the quality of proteins, we opt for entries from UniProtKB/Swiss-Prot, a highly reliable and manually annotated protein sequence database. We then augment our dataset by incorporating selected entries with high annotation scores from UniProtKB/TrEMBL, thus enhancing data volume and diversity. Further, to mitigate potential bias in training due to redundancy in protein sequence data, such as closely-related protein variants and homologous sequences with high sequence identity within Swiss-Prot and TrEMBL databases, we administer a stringent filtration process to purge redundant proteins from our dataset. First, We cluster these protein sequences based on a 90% similarity threshold using MMseqs (Steinegger & Söding, 2017) tool (–min-seq-id 0.9), designate entries abundant in functional annotations within each cluster as representative proteins and dismissing the remainder. Second, MMseqs search (–min-seq-id 0.9) is executed using the representative proteins sourced from the TrEMNL as query database, and proteins from the Swiss-Prot as the target database. All TrEMNL sequences that align with a Swiss-Prot sequence, possessing 90% sequence identity or higher during this search, are subsequently eliminated. These measures intend to streamline the construction of high-quality, non-redundant protein function-oriented instruction data.

**Protein design**    In this work, we seek to construct instructions using the textual form to implement *de novo* protein design for user intent, which could conveniently design composite properties of interest. Formally, given the design requirements of users, models are required to generate protein amino acid sequences that align with those requirements. Hence, we select annotations from 20 features in the UniProt knowledgebase to generate the protein design requirements as instructions. As shown in Table 6, these features comprise the properties of proteins commonly studied or expected for target proteins by biological researchers. To better integrate this structural knowledge with instructions, we design a series of templates (see in Table 7) to convert the annotations into textual expected properties. Therefore, we could assemble the above functional descriptions and properties together into the protein design instructions (we see an annotation as an individual condition). Table 7 demonstrates examples of templates for the conversion of structural data into textual format. Thus, we obtain 200,000 instruction entries, with the *input* being the design requirement and the *output* being the ideal protein sequence. In this task, the provided target sequence serves merely as a reference, which contrasts with other functional prediction tasks that have a definitive ground truth.

Over time, the UniProt database has amassed an extensive collection of experimentally verified functional descriptions and annotations for proteins. This collection assists researchers in swiftly comprehending the roles and mechanisms these proteins play in various biological processes. Nevertheless, the swift augmentation in protein numbers within sequence databases, coupled with the diversity of their functions, poses a significant challenge for automated function prediction through computational approaches. In our work, we focus on three distinct functional annotation tasks and

Table 6: Twenty annotation features for formulating precise constraints for protein design. When it comes to features that lack specific types such as signal, which simply indicates the presence of a particular feature in a given protein, we utilize '-' to represent this in the 'Example' field.

| Feature | Content | Example |
|---|---|---|
| Function | General function(s) of a protein | Tube-forming lipid transport protein which mediates the transfer of lipids between membranes at organelle contact site. |
| Pathway | Associated metabolic pathways | Nitrogen metabolism |
| Co-factor | Non-protein substance required for enzyme activity | $Ni^{2+}$ |
| Catalytic activity | Reaction(s) catalyzed by an enzyme | (S)-ureidoglycolate = glyoxylate + urea |
| Gene Ontology (MF) | Molecular level activities performed by proteins | ureidoglycolate lyase activity |
| Gene Ontology (BP) | The biological processes accomplished by multiple molecular activities | signal transduction |
| Gene Ontology (CC) | The locations relative to cellular structures in which the protein performs a function | mitochondrion |
| Signal | Sequence targeting proteins to the periplasmic space or secretory pathway | - |
| Coiled coil | Regions of coiled coil within the protein | - |
| Motif | Short sequence motif of biological interest | Nuclear localization signal |
| Domain | Type of each modular protein domain | PWWP domain |
| Compositional bias | Compositional bias in the protein | Polar residues |
| Topological domain | Non-membrane regions of membrane-spanning proteins | Lumenal, melanosome |
| Transmembrane | Extent of a membrane-spanning region | Helical transmembrane region |
| DNA binding | Type of a DNA-binding domain | Homeobox |
| Binding site | Binding site for any chemical group | ATP |
| Active site | Amino acid(s) directly contributing to the enzymatic activity | For GATase activity |
| Turn | Turns | - |
| Beta strand | Beta strand regions | - |
| Helix | Helical regions | - |

Table 7: Templates for converting structured annotations of specific protein fields into textual form. In practice, we also utilized LLM (GPT-3.5) to enrich the style of the templates.

| Feature | Template |
|---|---|
| Function | For general function, the protein need meet that {`function`}. |
| Pathway | A protein that is capable of catalyzing reactions within the {`pathway`} with high efficiency and specificity. |
| Co-factor | The protein must bind to {`co-factor`} in order to perform its enzymatic function. |
| Catalytic activity | The protein should be designed to catalyze the following reaction {`catalytic activity`}. |
| Gene Ontology (MF) | The protein must be able to {`molecular function`}. |
| Gene Ontology (BP) | The designed protein must be able to regulate {`biological process`}. |
| Gene Ontology (CC) | The designed protein localizes to the {`cellular component`}. |
| Signal | Incorporate a signal peptide in the protein design. |
| Coiled coil | The target protein must incorporate a coiled coil domain. |
| Motif | The protein's functional design necessitates the inclusion of a {`motif`}. |
| Domain | The designed protein should contain one or more {`domains`} that are essential for its function. |
| Compositional bias | The protein should exhibit {`compositional bias`}. |
| Topological domain | The {`topological domain`} of the protein should be designed to be flexible or rigid. |
| Transmembrane | The protein should contain a {`transmembrane`} membrane-spanning region. |
| DNA binding | A protein with {`DNA binding`} capability for targeted gene regulation. |
| Binding site | The protein should be able to bind {`binding site`} ligand in a variety of conditions |
| Active site | The designed protein must have a {`active site`} that is highly conserved among related enzymes. |
| Secondary structure | The target protein must exhibit {`Beta strand, Helix, Turn`} as its primary conformation. |

one task dedicated to generating functional descriptions. Our goal is to assess the ability of generative language models to address a broad spectrum of functional prediction issues under a unified framework. Concretely, we consider two widely used classification schemes that organize these myriad protein functions, Gene Ontology (GO) Consortium (Ashburner et al., 2000) and Enzyme Commission (EC) numbers.

**Protein function prediction**   For GO terms prediction, given the specific function prediction instruction and a protein sequence, models characterize the protein functions using the GO terms presented in three different domains (cellular component, biological process, and molecular function). In the acquired 116,458 instruction entries, the *input* is a protein sequence, and the *output* represents the function of that protein.

**Catalytic activity prediction**   Meanwhile, the EC number, a numerical classification system for enzymes hinging on the chemical reactions they catalyze, is substituted with the corresponding reaction. This substitution aims to leverage the tacit knowledge ingrained in pre-trained language models, thereby encouraging the model to predict the reaction itself rather than the mere EC number. In the final 54,259 instruction entries, the *input* is a protein sequence, and the *output* refers to the catalytic activity of the protein and the chemical reactions it promotes.

**Domain/motif prediction**   We introduce the domain prediction task, which tasks language models with the identification of the domain type within a given protein sequence, which is defined as a compact folded three-dimensional structure. Each of the 46,028 instruction entries includes a protein sequence as the *input*, and the *output* refers to the domains or motifs that the protein may contain.

**Functional description generation**   On the basis of the above functional prediction tasks, we design the functional description generation task, which not only evaluates the reasoning capability of the language model in determining the function of a protein sequence but also assesses the efficacy of the language model's text generation. After data preprocessing, we obtain 88,259 instruction entries, where the *input* is a protein sequence, and the *output* describes the protein's function, subcellular localization, and any biological processes it may be a part of.

## B.3   BIOMOLECULAR TEXT INSTRUCTIONS

**Chemical entity recognition**   Chemical Entity Recognition (CER) is a fundamental task in biomedical text mining and Natural Language Processing (NLP). It involves the identification and classification of chemical entities in textual data, such as scientific literature. These entities can encompass a broad range of concepts including chemical compounds, drugs, elements, ions or functional groups. Given the complexity and variety of chemical nomenclature, the CER task represents a significant challenge for LLMs, and their performance in this task can provide important insights into their overall capabilities in the biomedical domain. For this task, we employ the BC4CHEMD (Krallinger et al., 2015) dataset, in which the chemical entities have been manually identified and labeled by experts. To ensure a balanced representation of each task, and to equip the model with the ability to handle a wide array of tasks, we randomly sample 1,000 entries from the BC4CHEMD dataset.

**Chemical-protein interaction extraction**   We task language models with the nuanced role of annotating chemical-protein interactions. This endeavor seeks to explore the extent of biochemical and pharmacological knowledge encapsulated within these models. More specifically, the models are presented with excerpts from scientific literature and are required to not only identify distinct chemicals within the text but also to discern the specific nature of the interactions between them. This could involve, for instance, determining regulatory relationships between identified ligands and proteins. We utilize the ChemProt (Krallinger et al., 2017) dataset as the primary source and subsequently convert it into instruction data. In instances where multiple relationship triplets exist within a common scientific document, we consolidate them into a singular annotation separated and delineated by commas.

**Chemical-disease interaction extraction**   The goal of this task is to discern the relationships between chemicals and diseases from given medical literature, a concept known as chemical-induced disease (CID) relations. These CID relations are paramount in biomedical research, playing crucial roles in areas such as drug discovery, toxicology, and disease etiology. We utilize the BC5CDR Corpus (Li et al., 2016), which comprises 1,500 PubMed articles. These articles collectively feature 4,409 annotated chemicals, 5,818 diseases, and 3,116 chemical-disease interactions. Considering the analogous structure of the BC5CDR dataset to the ChemProt dataset, we employ an identical series of processing steps to convert the dataset entries to instructions.

---

**Prompt:**
Given a paragraph of paper, can you ask five questions and give answers that can be directed to the paragraph? List in the format of Question-Answer pair.

Paragraph: The outer membrane protein G (OmpG) nanopore is a monomeric $\beta$-barrel channel consisting of seven flexible extracellular loops. Its most flexible loop, loop 6, can be used to host high-affinity binding ligands for the capture of protein analytes, which induces characteristic current patterns for protein identification. At acidic pH, the ability of OmpG to detect protein analytes is hampered by its tendency toward the closed state, which renders the nanopore unable to reveal current signal changes induced by bound analytes. In this work, critical residues that control the pH-dependent gating of loop 6 were identified, and an OmpG nanopore that can stay predominantly open at a broad range of pHs was created by mutating these pH-sensitive residues. A short single-stranded DNA was chemically tethered to the pH-insensitive OmpG to demonstrate the utility of the OmpG nanopore for sensing complementary DNA and a DNA binding protein at an acidic pH.

---

**GPT-3.5 completion:**

Q1: What is the OmpG nanopore?
A1: The OmpG nanopore is a monomeric $\beta$-barrel channel consisting of seven flexible extracellular loops.

Q2: What is loop six used for?
A2: Loop 6 is used to host high-affinity binding ligands for the capture of protein analytes, which induces characteristic current patterns for protein identification.

Q3: How can the tendency toward the closed state be addressed?
A3: Critical residues that control the pH-dependent gating of loop 6 can be identified and an OmpG nanopore that can stay predominantly open at a broad range of pHs can be created by mutating these pH-sensitive residues.

Q4: How has the utility of the OmpG nanopore been demonstrated?
A4: The utility of the OmpG nanopore has been demonstrated by chemically tethering a short single-stranded DNA to the pH-insensitive OmpG to sense complementary DNA and a DNA binding protein at an acidic pH.

Q5: What type of channel is the OmpG nanopore?
A5: The OmpG nanopore is a monomeric $\beta$-barrel channel.

---

Figure 7: An illustration of SELF-QUESTIONING method for generating factual question-answer pairs based on a given PubMed abstract derived from PubMed White (2020) in the biomolecular field.

**Multiple-choice question**   We assemble a collection of 12,398 multiple-choice questions from the MedMCQA (Pal et al., 2022) and MMLU (Hendrycks et al., 2021) (Massive Multitask Language Understanding) datasets. Using the instruction-tuning approach, our objective is to inspire the professional capabilities of language models to answer diverse probes pertaining to biomolecules. The questions span the subjects in biology, chemistry, medicine and other vertical areas associated with biomolecular research.

**True or False question**   The objective of the true or false question-answer task is to answer research questions with affirmative, negative, or uncertain responses (e.g., Do preoperative statins reduce atrial fibrillation after coronary artery bypass grafting?). In the initial PubMedQA dataset, each entry consists of a question, an abstract from a publication in PubMed (White, 2020) for reference, a single answer and a corresponding explanation for the answer. For this task, instead of rigidly designing an instruction that prompts models to refer to the materials for answering, we eliminate the references and treat the questions as complements to the subsequent open-ended questions, accommodating a wider variety of question types. Further, the ideal response to a given question should provide an accurate answer with a cogent and reasonable explanation.

**Open question**   Open-ended questions are defined as those that simply pose the question, without imposing any constraints on the format of the response. This distinguishes them from questions with a predetermined answer format. We primarily gather the open-ended questions from the MedMCQA dataset, specifically selecting those pertaining to the domains of biochemistry, pharmacology and medicine. The original format of MedMCQA questions was a multiple-choice style, illustrated by questions such as, "In which of the following conditions are Leukotriene inhibitors highly effective?". However, we found that certain questions did not clearly communicate that the answers needed to be

Table 8: Data statistics of biomolecules.

| | Characteristics | Min | Max | Mean | Median |
|---|---|---|---|---|---|
| | Bertz complexity | 0 | 36,222 | 508 | 206 |
| | Molecular weight | 1 | 8,656 | 273 | 129 |
| | Atom count | 1 | 574 | 19 | 9 |
| | Ring count | 0 | 75 | 3 | 2 |
| | Sequence length | 2 | 39677 | 455 | 391 |
| | Domain | 1 | 2215 | 27 | 3 |
| | Catalytic activity | 1 | 2404 | 15 | 2 |
| | Gene Ontology | 1 | 28924 | 35 | 4 |

selected from the provided choices, for example, a question like "Antibody used in the treatment of RSV infection is:". Hence, we conveniently transform 27,574 questions into open-ended questions, where the corresponding answer encompasses the description of the correct option and the explanation, namely MedMCQA-Open.

In order to delve deeper into the queries concerning biomolecules and bolster the language models' proficiency within the domains of chemistry and biology, we propose the SELF-QUESTIONING to expand the instruction data of open-ended question by employing GPT3 to extract factual question-answer pairs from PubMed abstracts in the field of biomolecular research. The implementation of SELF-QUESTIONING consists of three steps: 1) data collection, 2) question-answer instance generation with the self-questioning approach, and 3) filtering low-quality questions.

First, we gather the complete abstracts from PubMed, the annual baseline released in December 2022. Each publication in PubMed comprises a title, abstract and Medical Subject Headings (MeSH). To focus on responding to questions about the biomolecular domain from users, we only consider those publications whose abstract and MeSH contain some specific keywords (e.g., protein, molecule) that probably pertain to the study of biomolecules. Second, it is challenging that expect models to generate high-quality and factual questions based on the provided reference materials. Nevertheless, we found that pretrained language models can achieve this when prompted with the instruction that requests the model to extract question-answer pairs. This approach ensures that the derived answers are properly aligned with the corresponding sources. A representative example of this generation process is illustrated in Figure 7. Third, to further ensure the overall quality of the produced questions, we incorporate an exclusion criterion that substantially eliminates questions closely tied to reference materials, for instance, questions such as "What is the purpose of the study?". These types of questions generally involve specific vocabulary terms such as "result", "study", "paragraph" and "article". By identifying such terms, we are able to exclude them from the pool of consideration effectively. We refer to the dataset containing 10,521 examples as PubMedQA-GPT. This is then combined with the MedMCQA-Open dataset to construct the instruction data.

## C A MORE EXHAUSTIVE DATA ANALYSIS MOL-INSTRUCTIONS

Figure 4 provides a multi-faceted analysis of the diversity and complexity of molecules and proteins. For the sake of clear presentation, only part of the coordinate coverage is displayed. A more comprehensive display of statistical data is provided in Table 8. Overall, these statistics reflect the broad and diverse nature of the biomolecules, which should contribute to the robustness and generalizability of models trained on them. Indeed, protein sequences are typically very long, which poses a significant challenge for LLMs to capture and understand their "grammar" or patterns. Strategies for encoding and processing these lengthy sequences, as well as effective ways to guide LLMs in predicting or designing valid protein sequences, are crucial areas for further investigation.

## D EXPERIMENTAL SETUP DETAILS

With the Mol-Instructions data, we employ the 7B LLaMA model as our foundation model in the entire experiment and conduct instruction tuning on the LLaMA model. In this work, due to the diversity of modalities and substantial variations in task difficulty, we train the LLaMA models on our three distinct datasets tailored to specific biomolecular problems, resulting in the development of three fine-tuned models. For each distinct task, we allocated almost 1k samples as the test set to

compute metrics and evaluate the model's performance. The remaining samples were divided into training and validation sets at an 8:2 ratio.

Although tokenization is a crucial part of data processing, particularly when dealing with various modalities, this study does not delve into an extensive analysis of the sequential tokenization of different modalities. Therefore, we approach the protein amino acid sequence and the SELFIES string of molecules as human language and tokenize the data using the byte-pair encoding (BPE) algorithm. We utilize the identical tokenization model employed in the LLaMA model. We use the longest-padding strategy to pad the tokenized sequences to the longest sequence length in the batch.

We adopt the low-rank Adapter (LoRA) fine-tuning on the molecule and text-oriented datasets, an efficient method that reduces memory during training and keeps a small set of parameters trainable, while not updating pre-trained models. For protein-oriented instruction data, we perform full finetuning with the memory optimization technique, ZeRO implemented in the DeepSpeed library. Our models are trained using AdamW optimizer and linear learning rate scheduler. We conduct the LoRA training and generation on 32GB V100 GPUs while performing the full-model finetuning on the 80GB A800 GPUs. The exact hyperparameters we tune for each model are shown in Table 9.

Table 9: Training hyperparameters for finetuning on different datasets. QV: two linear transformation matrices on the query and value states in the self-attention module.

| Hyperparameter | Molecule | Protein | Text |
|---|---|---|---|
| Finetune method | LoRA | Full | LoRA |
| Batch size | 800 | 96 | 1024 |
| LR | 3e-4 | 2e-5 | 3e-4 |
| Steps | 40,000 | 25,000 | 840 |
| Warmup steps | 1,000 | 2,500 | 100 |
| LoRA $r$ | 16 | - | 16 |
| LoRA $\alpha$ | 16 | - | 16 |
| LoRA dropout | 0.05 | - | 0.05 |
| LoRA layers | QV | - | QV |

# E   EVALUATION METRICS

## E.1   MOLECULE METRICS

To evaluate the molecular understanding tasks, we utilize metrics like BLEU (Papineni et al., 2002), ROUGE (Lin, 2004), and METEOR (Banerjee & Lavie, 2005) to assess the quality of the generated outputs by comparing them to reference answers. For molecule generation, we first validate whether the generated strings are valid molecules using RDKit (Landrum et al., 2013) and then compute their exact match with the reference solutions. However, it's important to note that a single textual description might correspond to multiple molecular structures. Moreover, expecting an LLM, even one fine-tuned with LoRA on specific instructions, to produce outputs that perfectly match reference data might be unrealistic. To accommodate these complexities and offer a more comprehensive evaluation, we further employ metrics that measure molecular similarity. These include similarity scores derived from RDKit/MACCS/Morgan fingerprints (Tanimoto, 1958; Schneider et al., 2015; Durant et al., 2002), as well as Levenshtein (Li & Liu, 2007) and BLEU scores. For the molecular property prediction task, we employ MAE (mean absolute error) to quantify how accurately the model predicts continuous values.

## E.2   PROTEIN METRICS

For protein understanding tasks, we esteem these tasks as the functional description task and employ ROUGE to quantify the quality of the generated descriptions for biological characteristics.

## E.3   BIOTEXT METRICS

For NLP text tasks, we employ general metrics for Q&A, entity recognition, and relation extraction to evaluate the generated outputs.

# F  ADDITIONAL RESULTS

Due to space constraints, only a portion of the results are displayed in the main text. The remaining results, which further illustrate the effectiveness of our Mol-Instructions, can be found in Figure 8, 9, 10, and 11. Please note that Galactica (Taylor et al., 2022), Text+Chem T5 (Christofidellis et al., 2023), and MolT5 (Edwards et al., 2022) only support the SMILES format. To accommodate this, we convert the SELFIES in our instructions to SMILES. It's also important to mention that MolT5 has not undergone instruction tuning, hence it only processes the "input" part of our instructions.

The analysis of the various experiments illustrated in Figure 8 reveals how Mol-Instructions guides LLMs in comprehending and executing specific molecular tasks. This guidance is evident in tasks such as molecule description generation and description-guided molecule design, where the dataset provides precise instructions that direct the model to produce detailed molecule descriptions and accurate designs. The performance of MolT5 is notably inferior to instruction-following models, exhibiting poor generalization capabilities. In tasks related to chemical reactions, such as forward reaction prediction, retrosynthesis, and reagent prediction, Mol-Instructions play a key role in instructing the model to predict reaction products, potential reactants, or correct reagents. Alpaca-LoRA, ChatGLM, Baize, and Vicuna in the absence of such specific instructions, produce outputs that are either too generic or not aligned with the task requirements. LLMs generally underperform in molecule generation compared to domain-specific smaller models, due to their broader focus which compromises task-specific specialization. For the property prediction tasks, Mol-Instructions help the model provide near-accurate estimations of various molecular properties such as HOMO, LUMO, and HOMO-LUMO gap energies. In comparison, Alpaca-LoRA's outputs, lacking guidance from specific instructions, deviate significantly from the actual values. Galactica consistently outputs negative values.

As demonstrated in the prediction instances illustrated in Figure 9 and 10, the fine-tuned model delivers consistent and user-specific protein analysis for functional annotation. For instance, the model accurately classifies a provided protein as the *23S rRNA (adenine(2503)-C(2))-methyltransferase* within the context of catalytic activity prediction tasks. Moreover, while slight variations may arise in describing distinct functional traits relative to the annotations in the UniProtKB, it showcases remarkable potential in discerning the fundamental properties of proteins.

As showcased in Figure 11, in tasks related to chemical entities and their interactions, Mol-Instructions aids the model in identifying the correct entities and establishing meaningful relations between them. In contrast, models that are not fine-tuned with our instructions, such as Alpaca-LoRA, often produce outputs that are too generic or miss the specific relationships present in the data. When dealing with multiple-choice, True or False, or open questions, the model fine-tuned with Mol-Instructions not only provides the correct answers but also delivers them in a more detailed and well-structured manner. This demonstrates the effectiveness of Mol-Instructions in improving LLMs' understanding and performance in these question-answering tasks.

Overall, the results indicate that Mol-Instructions can improve the LLM's ability to execute molecule-oriented, protein-oriented, and biomolecular text tasks. It underscores the importance of having task-specific instructions to guide the model, particularly when dealing with specialized domains such as biochemistry. In comparison, models that lack such specific instruction-based tuning, like Alpaca-LoRA, may struggle to produce outputs that align with the specific requirements of the tasks.

Despite the promising results demonstrated above, there are also several limitations. First, while the experiments confirm that Mol-Instructions can aid LLMs in understanding and mastering biomolecular-related information to a certain extent, it should be clear that the model obtained by instruction tuning at this stage is merely a preliminary demonstration. Its application potential for real-world, production-level tasks is still limited. Moreover, to fully realize the potential of Mol-Instructions, more profound exploration and innovation in algorithmic improvement are needed. This includes optimizing instruction tuning methods, further refining the model's ability to understand and follow instructions, and enhancing its generalization capacity across various molecular tasks. Therefore, while Mol-Instructions represents a step forward in the use of instruction datasets for biomolecular tasks, there remains a significant scope for future research and development to maximize its utility and effectiveness.

---

**(a) Molecule**

**Molecular description generation**

**Instruction:** Please give me some details about this molecule.

**Input:** [C][C][C][C][C][C][C][C][C][C][C][C][C][C][C][C][C][C][C][C][=Branch1][C][=O][O][C@H1][Branch2][Ring1][=Branch1][C][O][C][=Branch1][C][=O][C][C][C][C][C][C][C][C][C][C][C][C][C][C][C][C][C][C][C][C][O][P][=Branch1][C][=O][Branch1][C][O][O][C][C@@H1][Branch1][=Branch1][C][=Branch1][C][=O][O][N]

**Output (ground truth):** The molecule is a 3-sn-phosphatidyl-L-serine in which the phosphatidyl acyl groups at positions 1 and 2 are specified as stearoyl and arachidonoyl respectively. It is functionally related to an arachidonic acid and an octadecanoic acid.

**Output (ours):** The molecule is a 3-sn-phosphatidyl L-serine in which the phosphatidyl acyl groups at positions 1 and 2 are palmitoyl and stearoyl respectively. It has a role as a mouse metabolite. It derives from an octadecanoic acid and a hexadecanoic acid.

**Output (Alpaca):** [C][C][C][C][C][C][C][C][C][C][C][C][C][C][C][C][C][C][C][C][=Branch1][C][=O][O][C@H1][Branch2][Ring1][=Branch1][C][O][C][=Branch1][C][=O][C][C][C][C][C][C][C][C][C][C][C][C][C][C][C][C][C][O][P][=Branch1]

**Output (ChatGLM):** I'm sorry, but I cannot provide detailed information about a molecule that appears to contain a mix of carbon, hydrogen, oxygen, and nitrogen atoms, as well as various other chemical elements and functional groups.

**Output (Baize):** This is a molecular structure of the compound known as Cytosine, which is one of the four bases (A, G, C, and T) that make up the nucleotide of DNA and RNA. The molecule consists of a five-membered ring structure with a nitrogen atom in the center. It is a part of the pyrimidine family of nucleobases.

**Output (Vicuna):** The molecule you have provided is a carbohydrate molecule with a branched structure. It consists of a chain of carbon atoms with hydroxyl groups (OH) attached to some of the carbon atoms. The molecule also has a ring structure attached to one of the carbon atoms in the chain.

**Output (Galactica):** CCCCCCCCCCCCCCCCCCC(=O)OC[C@H1](COC(=O)CCCCCCCCCCCCCCC)COP(=O)(O)OC[C@@H1](C(=O)O)N[C@@H1](C(=O)O)COP(=O)(O)OC[C@@H1](C(=O)O)N[C@@H1](C(=O)O)COP(=O)(O)OC[C@@H1](C(=O)O)N[C@@H1](C(=O)O)COP(=O)(O)OC[C@@H1](C(=O)O)N[C@@H1](C(=O)O)COP(=O)

**Output (Text+Chem T5):** The molecule is a cyclic tetracyclic analogue of nicotinic acid. It has a role as an antineoplastic agent and a bacterial metabolite. It is a cyclic tetracyclic analogue and a tetracyclic analogue. It derives from a nicotinic acid.

**Output (MolT5):** The It It an an  an is an It an an an an an an an an an an an an an an an an an an an an an  an  an  an  an  an  an  an an an

**Forward reaction prediction**

**Instruction:** With the provided reactants and reagents, propose a potential product.

**Input:** [O][=N+1][Branch1][C][O-1][C][=C][N][=C][Branch1][C][Cl][C][Branch1][C][I][=C][Ring1][Branch2].[Fe]

**Output (ground truth):** [N][C][=C][N][=C][Branch1][C][Cl][C][Branch1][C][I][=C][Ring1][Branch2]

**Output (ours):** [N][C][=C][N][=C][Branch1][C][Cl][C][Branch1][C][I][=C][Ring1][Branch2]

**Output (Alpaca):** The potential product is a polycyclic aromatic hydrocarbon (PAH).

**Output (ChatGLM):** Based on the given reactants and reagents, a potential product could be [Fe][=N+1][Branch1][C][Cl][C]. The reaction between O, =N+1, Branch1, C, O-1, C, =C, N, =C, Branch1, C, Cl, C, I

**Output (Baize):** Based on the provided reactants and reagents, the potential product is Fe(CN)6. The provided reactants are a cytosine molecule (C), an oxygen molecule (O), and a nitrogen molecule (N). The reagent [Fe] is a metal atom, which can undergo a reaction with the cytosine molecule to form a complex compound.

**Output (Vicuna):** Based on the provided input, a potential product could be: [O][=N+1][Branch1][C][O-1][C][=C][N][=C][Branch1][C][Cl][C][Branch1][C][I][=C][Ring1][Branch2].[Fe]

**Output (Galactica):** C[C@H1][C@H]2[C@@H]3C[C@H2][C@H1][C@@H]3O

**Output (Text+Chem T5):** O=[N+]([O-])C1=CN=C(Cl)C(I)=C1

**Reagent prediction**

**Instruction:** Please provide possible reagents based on the following chemical reaction.

**Input:** [C][C][=C][C][=C][Branch1][C][N][C][=N][Ring1][#Branch1].[O][=C][Branch1][C][Cl][C][Cl]>>[C][C][=C][C][=C][Branch1][Branch2][N][C][=Branch1][C][=O][C][Cl][C][=N][Ring1][O]

**Output (ground truth):** [C][C][C][O][C][Ring1][Branch1].[C][C][N][Branch1][Ring1][C][C][C][C].[O]

**Output (ours):** [C][C][C][O][C][Ring1][Branch1].[C][C][N][Branch1][Ring1][C][C][C][C]

**Output (Alpaca):** Potential reagents that might have been utilized in the provided chemical reaction: [C][C][=C][C][=C][Branch1][C][N][C][=N][Ring1].

**Output (ChatGLM):** Here are some possible reagents based on the given chemical reaction: 1. Cinnamaldehyde (C2H5OH) 2. Nitrogen gas (N2) 3. Cinnamaldehyde + N2 → Cinnamamide (C2H5NO) 4. Nitrogen gas + HCl → H2N + Cl2 5. Nitrogen gas +NaOH → NaCN + H2O

**Output (Baize):** Based on the provided chemical reaction, the possible reagents are: A cytosine molecule (C), An oxygen molecule (O), A nitrogen molecule (N), A carbon molecule (C), An oxygen molecule (O), A chlorine molecule (Cl), A carbon molecule (C), A chlorine molecule (Cl).

**Output (Vicuna):** Based on the given chemical reaction, the possible reagents that could be used is: [C][C][=C][C][=C][Branch1][C][N][C][=N][Ring1][#Branch1].[O][=C][Branch1][C][Cl][C][=N][Ring1][O]

**Output (Galactica):**
COC1=CC=C(C(=O)NC2=CC=CC=C2C(=O)NC2=CC=CC=C2C(=O)NC2=CC=CC=C2C(=O)NC2=CC=CC=C2C(=O)NC2=CC=CC=C2C(=O)NC2=CC=CC=C2C(=O)NC2=CC=CC=C2C(=O)NC2=CC=CC=C2C(=O)NC2=CC=CC=C2C(=O)NC2=CC=CC=C2C(=O)NC2

**Output (Text+Chem T5):** CC1=CC=C(NC(=O)CCl)C=N1

**Description-guided molecule design**

**Instruction:** Create a molecule with the structure as the one described.

**Input:** The molecule is a primary arylamine in which an amino functional group is substituted for one of the benzene hydrogens. It is a primary arylamine and a member of anilines.

**Output (ground truth):** [N][C][=C][C][=C][C][=C][Ring1][=Branch1]

**Output (ours):** [N][C][=C][C][=C][C][=C][Ring1][=Branch1]

**Output (Alpaca):** The molecule is a primary arylamine in which an amino functional group is substituted for one of the benzene hydrogens. It is a primary arylamine and a member of anilines.

**Output (ChatGLM):** The molecule you described is called amino arylamine, also known as aniline. The structure of this molecule is as follows: C6H5NH2.

**Output (Baize):** The molecule would have the structure: C6H5NH2 (NH2 is the amino functional group).

**Output (Vicuna):** The molecule you are referring to is aniline (C6H5N). Aniline is a primary arylamine in which an amino functional group is substituted for one of the benzene hydrogens. It is a primary arylamine and a member of anilines.

**Output (Galactica):** CC(C)(C)C1=CC=C(C=C1)C=CCCNC(=O)OC(C)(C)C

**Output (Text+Chem T5):** C1=CC=C(C=C1)N

**Output (Mol T5):** C1=CC=C(C=C1)N

**Retrosynthesis**

**Instruction:** Please suggest potential reactants used in the synthesis of the provided product.

**Input:** [C][=C][C][C][N][C][=Branch1][C][=O][O][C][Branch1][C][C][Branch1][C][C][C]

**Output (ground truth):** [C][=C][C][C][N].[C][C][Branch1][C][C][Branch1][C][C][O][C][=Branch1][C][=O][O][C][=Branch1][C][=O][O][C][Branch1][C][C][C][Branch1][C][C][C]

**Output (ours):** [C][=C][C][C][N].[C][C][Branch1][C][C][Branch1][C][C][O][C][=Branch1][C][=O][O][C][=Branch1][C][=O][O][C][Branch1][C][C][C][C]

**Output (Alpaca):** The potential reactants used in the synthesis of the provided product are acetylene, carbon monoxide, and cyanide.

**Output (ChatGLM):** The product you have provided appears to be the amideamide bond, which is formed by the reaction of an amide group (=C-NH2) with a base such as sodium amide or lithium amide. Here are some potential reactants that could be used in the synthesis of the amideamide bond: 1. N-methylamide 2. N-methylmorpholine 3. C6H14N2O2 4. C2H5NO2 5. C2H5NH2

**Output (Baize):** Based on the provided chemical structure, the potential reactants that can be used in the synthesis of the product are: 1. A cytosine molecule (C) 2. An oxygen molecule (O) 3. A nitrogen molecule (N) 4. A carbon molecule (C) 5. An oxygen molecule (O) 6. A chlorine molecule (Cl)

**Output (Vicuna):** Based on the structure provided, potential reactants for the synthesis of the product could include: 1. Sodium metal (Na) - This could be used as a source of sodium ions (Na+) to react with the carbonyl group (C=O) present in the structure. 2. Ammonia (NH3) - This could be used as a source of nitrogen atoms (N) to react with the carbonyl group (C=O) present in the structure.

**Output (Galactica):** C[C@H1]C[C@@H]2[C@H]1[C@@H]1O[C@@H]12

**Output (Text+Chem T5):** C=CCCNC(=O)OC(C)C

**Property prediction**

**Instruction:** Please provide the HOMO energy value for this molecule. / Please provide the lowest unoccupied molecular orbital (LUMO) energy of this molecule. / I need to know the HOMO-LUMO gap energy of this molecule, could you please provide it?

**Input:** [C][C][O][C][C][Branch1][C][C][C][Branch1][C][C][C][C]

**Output (ground truth):** -0.2482 / 0.0844 / 0.3326

**Output (ours):** -0.2511 / 0.0835 / 0.3323

**Output (Alpaca):** 5.44 / -5.04 / 5.04

**Output (ChatGLM):** Without more information about the structure of the molecule, it is not possible to provide the exact HOMO/LUMO/HOMO-LUMO energy.

**Output (Baize):** -0.44 / 0.44 / 0

**Output (Vicuna):** 4.2/-3.2/4.5

**Output (Galactica):** -0.5983/-0.0231/-0.0231

Figure 8: Performance comparison of LLM tuning using instructions across **molecule-oriented** tasks.

---

**(b1) Protein**

**Protein function prediction**

**Instruction:** Analyze the following amino acid sequence, and determine the function of the resulting protein, its subcellular localization, and any biological processes it may be part of:

**Input:**
MNGTVNASAPSKMSEVAVERLSNDKALKVIFVLGGPGSGKGTQCAKIAKHFGFTHLSVGDLLRAEINSGSKNGTMIESMINEGKIVRSEVTIKLLQRAMHES
GNDKFLIDGFPRNEENRAAFENLEKIEPEFVLFFDCPMEEMERRILNRNQGRDDDKMETIRKRFKVFIESTLPVIEFYNLKGKLYKIDACKPADEVFEDVKAIF
SRFRAKEDSSQQTNICTAKRFELVMCLIKRLFREIKRMWSSFFCKAL

**Output (ground truth):** The protein characterized by the amino acid sequence demonstrates ATP binding, cytidylate kinase activity, uridylate kinase activity and is implicated in the 'de novo' pyrimidine nucleobase biosynthetic process, phosphorylation, pyrimidine nucleotide biosynthetic process. Its subcellular localization is primarily within the cytoplasm, nucleus.

**Output (ours):** Based on the given amino acid sequence, the protein appears to have a primary function of ATP binding, cytidylate kinase activity, UMP kinase activity. It is likely involved in the 'de novo' pyrimidine nucleobaseline cyclization, pyrimidine ribonucleoside biosynthetic process, uridine metabolic process, UMP catabolic process, and its subcellular localization is within the cytoplasm.

**Output (Alpaca):** The amino acid sequence
MNGTVNASAPSKMSEVAVERLSNDKALKVIFVLGGPGSGKGTQCAKIAKHFGFTHLSVGDLLRAEINSGSKNGTMIESMINEGKIVRSEVTIKLLQRAMHES
GNDKFLIDGFPRNEENRAAFENLEKIEPEFVLFFDCPMEEMERRILNRNQGRDDDKMETIRKRFKVFIESTLPVIEFYNLKGKLYKID

**Output (ChatGLM):** To determine the function of the resulting protein, we need to analyze its amino acid sequence and predict its molecular features, such as its structure, functionality, and interactions with other molecules. Function: The protein you provided is called 'Hsp70', which is a stress-responsive protein that is widely distributed in various organisms.

**Output (Baize):** This amino acid sequence does not correspond to a known protein. It is an artificial sequence that has not been characterized in any way. Therefore, the function, subcellular localization, and biological processes of the resulting protein cannot be determined.

**Output (Vicuna):** Based on the amino acid sequence provided, the resulting protein is likely to be a membrane-associated protein with a signal peptide at the N-terminus and a transmembrane domain in the middle. The protein is predicted to be localized in the endoplasmic reticulum (ER) and may be involved in lipid metabolism or transport. The C-terminal region of the protein may be involved in protein-protein interactions or signaling pathways related to lipid metabolism or transport. Further experimental validation is needed to confirm the function and localization of this protein.

**Output (Galactica):** ## Summary **Protein**: Probable serine/threonine-protein kinase 1; **Gene**:STK1; **Organism**: Homo sapiens (Human); **Status**: evidence at protein level; ## Function ### Catalytic Activity - ATP + L-seryl-[protein] = ADP + H(+) + O-phospho-L-seryl-[protein] - ATP + L-threonyl-[protein] = ADP + H(+) + O-phospho-L-threonyl-[protein]

**Protein design**

**Instruction:** Design a protein sequence that exhibits the desired activity and specificity.

**Input:**
1. The presence of Mg(2+) is necessary for the protein to function in the desired environment.
2. The AMP, (6S)-NADPHX binding site should be located in a region of the protein that is accessible to the ligand.
3. The designed protein should have ATP binding, NADPHX epimerase activity, metal ion binding, ADP-dependent NAD(P)H-hydrate dehydratase activity to facilitate nicotinamide nucleotide metabolic process.
4. For general function, the protein need meet that Catalyzes the epimerization of the S- and R-forms of NAD(P)HX, a damaged form of NAD(P)H that is a result of enzymatic or heat-dependent hydration

**Output (BLAST):**
(Target Protein Accession: A0A518LQL6)
MSNELVLSREQVRRVDQRAIEAYGVPGIVLMENAGRGAAEIIRAACPSAQRVLIACGPGNNGGDGFVIARHLANAGWMVELLLACPADRITGDAQGNHEII
RRMNLPCAVMADARDLEAANDRFATADVIVDALLGTGASGPPREPIASLIRAINEAHRRVSAQPAPSVFAVDIPSGLDCDTGEAANPTVRADHTITFVARKIGF
RNPAARDLLGRVHVVDIGAPRAAIQDALTGKSG

**Output (ours):** Here's a protein sequence with the above functions:
MSYKIAVIGVTGSIAAFKSCELLRDTFQAGHEVILILSTESARDFIPMTALTRGATMIDLSFTSESVEWVDVPDEISNGTWHSGEVSMAAAKGLCDIFIAPASAN
IIAKFAHLGAIVSDTVLACTTPCLMLAPVMIEAMRCHSKETFLGRGVSPDLVEIRRILDSSKGILITSGNTLEEVSYEDIAEGIKKLPLTGTLKGSAPIELPLTHN
RSHVFSCGGNEKAEKLRERFGSRLINPTSRFDVCFFEPDKLYSLKEIFKGFRDYEEWSNSPEEIWRKKTEEVARELGYVDDKVVIICGNNNAGRAALLACRY
KPSRSENGFWKGFSGNPRGPYYDPDSWEEQKKVMEEFPKLVVCDTAALTSFEVGKRFRTNDIQLPAGVALASKLGCYPGHLPDAEYPQVPPKPFDNLHPVV
LFPPNR

**Sequence Alignment:**

(target)  MSNELVLSREQVRRVDQRAIEAYGVPGIVLMENAGRGAAEIIRAACPSAQRVLIACGPGNNGGDGFVIARHLANAG-WMVELLLACPADRITGDAQGNHEI   100
 (ours)   --MSKVLTPQEMREIDRYAIEEIGIPSVMLMENAGRGAAEVIAERWPSGEFYIFFCGPGNNGGDGLVIARHLANWSGPVAALFLGNPEAKLRGVSPDFTNE

(target)  IRRMNLPCAVMADARDLEAANDRFATADVIVDALLGTGASGPPREPIASLIRAINEAHRRVSAQPAPSVFAVDIPSGLDCDTGEAANPTVRADHTITFVA   200
 (ours)   IKRLNLPIQLLEEG---------LIVDCIVDAILGTGFTGGPLRTTWELVEAVNNASLAISLPVPDDVFDIGFPPTTVSFDLSIL---DLSDEAFSLVS

**Output (Alpaca):** The designed protein sequence should contain the following amino acids:
Mg(2+) binding site: Gly-Gly-Gly-Gly-Gly-Gly-Gly-Gly-Gly-Gly-Gly-Gly-Gly-Gly-Gly-Gly-Gly-Gly-Gly-Gly-Gly-Gly-Gly-Gly-Gly-Gly-Gly-Gly-Gly-Gly-
Gly-Gly-Gly-Gly

**Output (ChatGLM):**
AAXAAAGYSAGAAAGYSAGAAAGYSAGAAAGYSAGAAAGYSAGAAAGYSAGAAAGYSAGAAAGYSAGAAAGYSAGAAAGYSAGAAAGYSAGAAAGYSAGAAA
GYSAGAAAGYSAGAAAGYSAGAAAGYSAGAAAGYSAGAAAGYSAGAAAGYSAGAAAGYSAGAAAGYSAGAAAGYSAGAAAGYSAGAAAGYSAGAAAGYSAGA

**Output (Baize):** The protein sequence should contain a region that binds Mg(2+) with high affinity, such as a metal-binding domain. This domain could be taken from a protein that is known to bind Mg(2+) in a similar environment.2. The protein should contain a region that binds the ligand, (6S)-NADPHX. This region could be designed using computer-assisted molecular modeling techniques, taking into account the known structure of related ligands.

**Output (Vicuna):** Based on the provided input, I have designed a protein sequence that meets the desired activity and specificity. The protein is composed of 300 amino acids and has the following features:
1. The presence of Mg(2+) is necessary for the protein to function in the desired environment.
2. The AMP, (6S)-NADPHX binding site is located in a region of the protein that is accessible to the ligand.

**Output (Galactica):** The protein should be able to bind to AMP, ADP, ATP, (6S)-NADPHX, NAD(P)HX, and Mg(2+). [IMAGE] Figure Caption: Figure 14.3.2: The proposed reaction catalyzed by the designed protein.

Figure 9: Performance comparison of LLM tuning using instructions across **protein-oriented** tasks. (1/2)

# G  LIMITATIONS AND ETHICAL CONCERNS

The advancements in LLMs and their applications in the realm of biosciences, as showcased in our study, undeniably open avenues for a multitude of beneficial applications. However, with great power comes great responsibility. We recognize the necessity to discuss potential limitations and ethical concerns associated with such advancements.

## G.1  POTENTIAL MISUSE IN BIOENGINEERING

The capability of LLMs to "forecast chemical reactions, architect new molecules as per instructions, and apprehend molecular structural attributes and reactive features" inevitably raises concerns. In the wrong hands, these capabilities might enable malicious actors to exploit the system for nefarious purposes, such as the synthesis of harmful biochemical agents or bioweapons.

—— **(b2) Protein** ——

**Domain/motif prediction**

**Instruction:** Given this protein sequence, can you identify any common protein motifs or domains that it may contain?
**Input:**
MANTKYIFITGGVVSSLGKGIAAASIGALLESRGLSVSLIKVDPYINVDPGTMSPFQHGEVFVTEDGTETDLDLGHYERFVRFKASKKNNFTAGKVYETVIRN ERKGNYLGGTVQVIPHITNEIKKRIKKGGQNKDIAIVEVGGTVGDIESQPFVEALRQMALELPNSSWAFVHLTLVPFINASGELKTKPTQHSVKELRSLGISPD VLVCRSEQELPKDEKNKIALFCSVPAKSVISMHDVDTVYSIPILLNKQKVDDTILKKLNLKIKKPNLNDWKRVVKAKLLPEKEVNVSFVGKYTELKDSYKSI NEALEHAGIQNKAKVNINFVEAEQITSQNVRKVLKKSDAILVPGGFGERGIEGMILACKYARENNVPYLGICLGMQIAIIEYARNVLKLKSANSTEFDSSTKFP VIGLITEWSDISGKKEKRTKNSDLGGTMRLGGQVCKLKKKSNSYKMYKKSEIIERHRHRYEVNPNYKDKMIEQGLDVVGTSIDGKLVEMIELPSHKWFLAC QFHPEFTSNPRDGHPIFNSYIKSTITK

**Output (ground truth):** Our predictive analysis of the given protein sequence reveals possible domains or motifs. These include: Glutamine amidotransferase, CTP synthase N-terminal domains.
**Output (ours):** Based on computational analysis, the provided sequence potentially contains the following protein domains or motifs: Glutamine amidotransferase type-1 domains.
**Output (Alpaca):** The protein sequence contains multiple protein motifs and domains. It contains an N-terminal signal peptide, a proline-rich region, a coiled-coil domain, a leucine-rich repeat domain, a proline-rich region.
**Output (ChatGLM):** 与艺术有关，具体是指与音乐、艺术、文学或文学有关
**Output (Baize):** Here are some of the possible motifs and domains: Glycine-rich region, Leucine-rich region, Kinase domain, Fibrinogen-like domain, Leucine-rich repeats, GTPase domain.
**Output (Vicuna):** Based on the provided protein sequence, there are several common protein motifs and domains that can be identified. These include: A signal peptide, which is indicated by the presence of the sequence "MANTKYIFITGGVVSSLGKGIAAASIGALLESRGLSVSLIKVDPYINVDPGTMSPFQHGEVFVTEDGTETDLDLGHYERFVRFKASKKNNFTAGKVYETVIR NERKGNYLGGTVQVIPHITNEIKKRIK"
**Output (Galactica):** ## Summary **Protein**: 10 kDa chaperonin; **Gene**: groS; **Organism**: Streptococcus pyogenes; **Status**: inferred from homology; ## Function What is the function of the protein 10 kDa chaperonin? binds to Cpn60 in the presence of Mg-ATP and suppresses the ATPase activity of the latter.

**Functional description generation**

**Instruction:** Examine the given protein sequence and share a brief overview of its attributes:
**Input:**
MKIVLASNNQGKLAELKAMLAPLGVQLLRQAELGIPEAAEPFRTFVENALAKARHASALSGLPALADDAGLCVEAFGGLPGVDTAFYAVQFGYAKGDANN VKALLAQMAGITDRRAALVSTLVAVRSAEDPEPLIACGRVAGEVALEPMGSNGFGFDPVMFIPEFGQTFAQLPVEVKNANSHRGKATRQMMALMRERWIT

**Output (ground truth):** A concise description of the protein with the specified amino acid sequence includes: Pyrophosphatase that catalyzes the hydrolysis of nucleoside triphosphates to their monophosphate derivatives, with a high preference for the non-canonical purine nucleotides XTP (xanthosine triphosphate), dITP (deoxyinosine triphosphate) and ITP. Seems to function as a house-cleaning enzyme that removes non-canonical purine nucleotides from the nucleotide pool, thus preventing their incorporation into DNA/RNA and avoiding chromosomal lesions.
**Output (ours):** Here is a summary of the protein with the given amino acid sequence: Pyrophosphatase that catalyzes the hydrolysis of nucleoside triphosphates to their monophosphate derivatives, with a high preference for the non-canonical purin nucleotides XTP (xanthosine triphosphate), dITP (deoxyinosine triphosphate) and ITP. Seems to function as a house-cleaning enzyme that removes non-canonical Purine nucleotides from the nucleotide pool, thus preventing their incorporation into DNA/RNA and avoiding chromosomal lesions.
**Output (Alpaca):** The protein sequence MKIVLASNNQGKLAELKAMLAPLGVQLLRQAELGIPEAAEPFRTFVENALAKARHASALSGLPALADDAGLCVEAFGGLPGVDTAFYAVQFGYAKGDANN VKALLAQMAGITDRRAALVSTLVAVRSAEDPEPLIACGRVAGEVALEPMGSNGFGFDPVMFIPEFGQTFAQLPVEVKNANSHRGKATRQMMAL
**Output (ChatGLM):** The given protein sequence is not provided. However, based on the amino acid sequence of a protein that was provided earlier, it appears to be a protein of unknown function.
**Output (Baize):** The given protein sequence is an artificial sequence that has not been characterized in any way. Therefore, its attributes cannot be evaluated or understood in any way. This sequence does not correspond to a known protein, and its function, subcellular localization, and biological processes cannot be determined.
**Output (Vicuna):** The given protein sequence appears to be a random assortment of amino acids without any discernible pattern or structure. A more detailed analysis would be necessary to determine any potential attributes or functions of this protein.
**Output (Galactica):** Multifunctional enzyme that displays cysteine protease, NTPase, RNA triphosphatase and RNA helicase activities. May play a role in the packaging and maturation of the virion. Specifically cleaves the capsid protein at two distinct sites. This protein has RNA-binding and NTPase activities and associates with the HBV RNA genome to promote encapsidation and replication. Also possesses a guanylyltransferase activity that is involved in maturation of the virion. Together, these activities participate in the packaging of the three reverse-transcribed copies of the genome of the HBV virion. Also plays a role in the biosynthesis of the virion envelope glycoproteins, in the budding of the virions through interactions with the cytoplasmic tail of the glycoprotein E2 at the ER membrane.

**Catalytic activity prediction**

**Instruction:** Please evaluate the following protein sequence and provide an explanation of the enzyme's catalytic activity, including the chemical reaction it facilitates:
**Input:**
MDKVAVAGFLPEELCASLSLSPSFRGNQIFQWIGKGVDSFDAMTNLSAELRASLAEKAILRSTRVSDVLKADDGTVKLQIQTEDDLAVETVLLTDKAARKTA CVSCQAGCAMGCAFCKTGTLGLARNLSAAEIVEQFLYLEKHAGALDNIVFMGMGEPLLNLDALRKAIAVLTDKRGRNLSSRRITVSTVGIVSGIYDLANNG PDVRLAVSLTTADETLRRELMPASLTNPLSDLRQAISYYIEKTGKRVTLEAVLLSGKNTSEKNADSLIAFAKGLDVHVNLIPWNPVEGLSFVTPDPEETAQFVS RLEKGGLNVTLRMHRGKSISGACGQLGKTNPYA

**Output (ground truth):** Based on the provided protein sequence, the enzyme appears to facilitate the chemical reaction: adenosine(37) in tRNA + 2 reduced [2Fe-2S]-[ferredoxin] + 2 S- adenosyl-L-methionine = 2-methyladenosine(37) in tRNA + 5'-  deoxyadenosine + L-methionine + 2 oxidized [2Fe-2S]-[ferredoxin] + S- adenosyl-L-homocysteine.
**Output (ours):** An analysis of the protein sequence reveals that the enzyme's catalytic function corresponds to the chemical reaction: adenosine(37) in tRNA + 2 reduced [2Fe-2S]-[ferredoxin] + 2 S-adenosyl-L-methionine = 2-methyladenosine(37) in tRNA + 5'- deoxyadenosine + L-methionine + 2 oxidized [2Fe-2S]-[ferredoxin] + S-adenosyl-L-homocysteine.
**Output (Alpaca):** This protein sequence is the amino acid sequence of the enzyme catalase. Catalase is an enzyme that catalyzes the breakdown of hydrogen peroxide into water and oxygen. The chemical reaction catalyzed by catalase is the decomposition of hydrogen peroxide into water and oxygen: $2H_2O_2 \rightarrow 2H_2O + O_2$
**Output (ChatGLM):** The protein sequence provided is the catalytic domain of a hypothetical enzyme, named "MLVAVAGFLPEELCASLSLSPSFRGNQIFQWIGKGVDSFDAMT". The catalytic domain contains a large loop ( loops 1-6 ) that is involved in the binding of the substrate and the formation of the catalytic site. The loop is surrounded by small amino acid residues that are involved in specific interactions with the substrate, such as hydrogen bonding and electrostatic interactions.
**Output (Baize):** This protein sequence corresponds to the active site of a class of enzymes known as cysteine protease, specifically, a member of the subfamily of cysteine proteases known as serine proteases. The amino acid sequence contains a conserved catalytic triad of residues (Cys, His, Asp) that are essential for the activity of the enzyme.The chemical reaction facilitated by the enzyme is the cleavage of a peptide bond between two amino acids by a serine protease. The specificity of the enzyme for the peptide bond that it cleaves is determined by the amino acid sequence of the substrate.
**Output (Vicuna):** The protein sequence provided is for an enzyme called pepsin, which is a proteolytic enzyme that catalyzes the hydrolysis of peptide bonds in protein molecules. The chemical reaction that pepsin facilitates is the breakdown of proteins into smaller peptides and individual amino acids. This reaction is an important step in digestion, as proteins are broken down into smaller molecules that can be more easily absorbed by the body.
**Output (Galactica):** ## Function Q: What is the function of the protein N-acetyl-gamma-glutamyl-phosphate reductase? Catalyzes the NADPH-dependent reduction of N-acetyl-5-glutamyl phosphate to yield N-acetyl-L-glutamate 5-semialdehyde. ### Catalytic Activity - N-acetyl-L-glutamate 5-semialdehyde + NADP(+) + phosphate = H(+) + N-acetyl-L-glutamyl 5-phosphate + NADPH

Figure 10: Performance comparison of LLM tuning using instructions across **protein-oriented** tasks. (2/2)

## G.2 Ethical Considerations in Generative AI

Recent research (Service, 2023; Sandbrink, 2023) have shed light on the potential misuse of generative AI in bioengineering. Our work, while focused on beneficial outcomes, does not turn a blind eye to these concerns. We believe that any generative AI tool, when made available to the public, should come with stringent usage guidelines and monitoring mechanisms to detect and prevent misuse.

(c) Biomolecular Text

**Chemical entity recognition**

**Instruction:** Find and list all the instances of the chemical entities in the following content.
**Input:** Both the control and caramiphen groups with double cannulas had significantly shorter latencies to seizure onset than the corresponding groups with single cannula.

**Output (ground truth):** caramiphen
**Output (ours):** caramiphen
**Output (Alpaca):** Both the control and caramiphen groups with double cannulas had significantly shorter latencies to seizure onset than the corresponding groups with single cannula.
**Output (ChatGLM):** Caramiphen is a drug, and cannula is a type of drug container.
**Output (Baize):** control group, caramiphen group, double cannula, latency to seizure onset, seizure, onset.
**Output (Vicuna):** Both the control and caramiphen groups with double cannulas had significantly shorter latencies to seizure onset than the corresponding groups with single cannula.
**Output (Galactica):** 1. 1. 2. 3. 4. 5. 6. 7. 8. 9. 10. 11. 12. 13. 14. 15. 16. 17. 18. 19. 20. 21. 22. 23. 24. 25. 26. 27. 28. 29. 30. 31. 32. 33. 34. 35. 36. 37. 38. 39. 40. 41. 42. 43. 44. 45. 46. 47. 48. 49. 50. 51. 52. 53. 54...
**Output (PMC-LLama):** The use of the double cannula significantly facilitated the monitoring of the experimental rats during the seizure tests. The ease with which experimental animals with double cannulated can be handled during seizure testing thus makes this the preferred method of cannulation.

**Chemical-disease interaction extraction**

**Instruction:** You are provided with a set of clinical trial summaries. Extract the chemical-disease relations from the summaries and present your findings in the format of (Subject, Object).
**Input:** Eating disorders and the associated behavioural problems and drug abuse are uncommon in pregnancy. When they do occur they are often unrecognized because of denial but when significant may pose a risk to both the mother and her fetus. This case illustrates a number of problems that may be encountered in women with eating disorders in pregnancy, including prolonged and recurrent metabolic disturbances and diuretic abuse. In particular it illustrates the derangements of thyroid function seen in pregnant women with eating disorders and reminds us that when a cause for thyrotoxicosis remains obscure, thyroxine abuse should be considered and explored.

**Output (ground truth):** (thyroxine, thyrotoxicosis)
**Output (ours):** (thyroxine, thyrotoxicosis)
**Output (Alpaca):** Eating disorders, Behavioural problems, Drug abuse (Subject, Object)
**Output (ChatGLM):** (Eating disorders, Behavioral problems, drug abuse, Eating disorders, Behavioral problems, drug abuse, Pregnancy, Eating disorders, Behavioral problems, drug abuse, Pregnancy, Metabolic disturbances, diuretic abuse, Pregnancy, Eating disorders, Behavioral problems, drug abuse, Thyroxine function, Pregnancy, Eating disorders, Behavioral problems, drug abuse, thyrotoxicosis)
**Output (Baize):** Subject: Pregnancy, Eating Disorders. Object: Metabolic Disturbances, Drug Abuse, Thyroid Function, Prolonged Thyrotoxicosis.
**Output (Vicuna):** Eating disorders - Pregnancy
**Output (Galactica):** (Mother, Thyroxine)
**Output (PMC-LLama):** yes

**Multi-choice question**

**Instruction:** Which of the following conditions does not show multifactorial inheritance?
**Input:** What of the following is the right choice? (A) Pyloric stenosis (B) Schizophrenia (C) Spina bifida (neural tube defects) (D) Marfan syndrome

**Output (ground truth):** The final answer is (D).
**Output (ours):** The final answer is (D).
**Output (Alpaca):** C) Spina bifida (neural tube defects)
**Output (ChatGLM):** Pyloric stenosis does not show multifactorial inheritance. It is a condition caused by a narrowing of the stomach outlet, which can lead to discomfort, nausea, and vomiting. The condition is inherited in an allelic fashion, meaning that it is caused by the presence of two different genetic variations.
**Output (Baize):** The correct answer is (D) Marfan syndrome.
**Output (Vicuna):** The condition that does not show multifactorial inheritance is (C) Spina bifida (neural tube defects). Spina bifida is a condition that occurs due to the failure of the neural tube to close properly during embryonic development, which is a single-gene defect. Pyloric stenosis, schizophrenia, and Marfan syndrome are all examples of multifactorial inheritance, which means they are caused by a combination of genetic and environmental factors.
**Output (Galactica):** (C) Spina bifida (neural tube defects) (D) Marfan syndrome
**Output (PMC-LLama):** OPTION D IS CORRECT.

**True or False question**

**Instruction:** Does feeding tube insertion and its timing improve survival?
**Input:** None

**Output (ground truth):** No, neither insertion of PEG tubes nor timing of insertion affect survival.
**Output (ours):** No, feeding tube insertion and its timing does not improve survival.
**Output (Alpaca):** Yes, feeding tube insertion has been shown to improve survival in critically ill patients.
**Output (ChatGLM):** Yes, feeding tube insertion and its timing can improve survival in certain situations.
**Output (Baize):** While feeding tube insertion and its timing may be beneficial in some cases, the relationship between feeding tube insertion and survival is complex and requires further research and understanding.
**Output (Vicuna):** Feed.
**Output (Galactica):** No
**Output (PMC-LLama):** no

**Chemical-protein interaction extraction**

**Instruction:** Your task is to extract the subject and object entities from the provided text and establish the corresponding relation triples. Please format your answers as (Subject, Relation, Object).
**Input:** Opioid receptors are the membrane proteins that mediate the pain-relieving effect of opioid drugs, such as morphine and fentanyl as well as endogenous opioid peptides enkephalins and endorphins. Using cDNAs for the mu and the kappa opioid receptors, we mapped the chromosomal locations of their genes in mouse. Multilocus cross analysis located the mu receptor gene Oprm on Chr 10 and the kappa receptor gene Oprk1 on Chr 1. Both genes are near centromere, with no markers more centromeric. These data indicate that the two opioid receptors are different gene products, ruling out the possibility that they may be differential splicing products from the same gene.

**Output (ground truth):** (morphine, regulator, Opioid receptors), (fentanyl, regulator, Opioid receptors)
**Output (ours):** (fentanyl, regulator, Opioid receptors), (morphine, regulator, Opioid receptors)
**Output (Alpaca):** (Subject, Relation, Object)(mu receptor, is_near, Chr 10)(kappa receptor, is_near, Chr 1)
**Output (ChatGLM):** (opioid receptors, membrane proteins, mediate pain-relieving effect ; mu and kappa opioid receptors, genes, mapped chromosome locations ; mu receptor gene Oprm on Chr 10 ; kappa receptor gene Oprk1 on Chr 1 ; Multi locus cross analysis, location ; centromere, mouse ; no markers more centromeric) (opioid receptors, gene products ; Morphine ; Endogenous opioid peptides enkephalins and endorphins)
**Output (Baize):** (Subject, Relation, Object) Subject: Opioid Receptors, Relation: Mediate, Object: Pain-Relieving Effect of Opioid Drugs and Endogenous Opioid Peptides.
**Output (Vicuna):** (Opioid receptors, are, the)
**Output (Galactica):** Oprm
**Output (PMC-LLama):** The mu opioid receptor gene Oprm locates to the distal region of Chromosome 10 in the mouse.

**Open question**

**Instruction:** What virus causes chronic disease?
**Input:** None

**Output (ground truth):** Hepatitis C virus (HCV).
**Output (ours):** Hepatitis C. Hepatitis C is a chronic liver disease caused by the hepatitis C virus (HCV).
**Output (Alpaca):** Hepatitis B is a virus that causes chronic liver disease.
**Output (ChatGLM):** There are many viruses that can cause chronic diseases. Some examples include: Chronic obstructive pulmonary disease (COPD), Chronic kidney disease, Chronic heart disease, Chronic diabetes, Chronic fatigue syndrome.
**Output (Baize):** There are several viruses that can cause chronic diseases, including: Hepatitis B and C, HIV, Herpes Simplex Virus (HSV), Human T-cell Leukemia Virus 1 (HTLV-1), Varicella-Zoster Virus (VZV).
**Output (Vicuna):** The virus that causes chronic disease is the Epstein-Barr virus (EBV).
**Output (Galactica):** Hepatitis C
**Output (PMC-LLama):** OPTION D IS CORRECT.

Figure 11: Performance comparison of LLM tuning using instructions across **biomolecular text** tasks.

### G.3 MITIGATING RISKS

Addressing the potential dangers is paramount. Some steps that can be taken include:

- *Regulated access*: Limiting access to only verified research entities or individuals with a track record of ethical research.
- *Monitoring usage patterns*: Implementing algorithms to detect patterns of misuse or exploration into potentially harmful domains.
- *Community oversight*: Establishing a community-driven oversight mechanism where experts can review and approve specific uses or generated outcomes.
- *Transparent reporting*: Encouraging users to report any unintended outcomes or potentially harmful use-cases they encounter.

The ethical implications of AI, especially in sensitive domains like biosciences, cannot be stressed enough. While our work represents a step forward in harnessing the capabilities of LLMs for good, it is vital to move forward with caution, diligence, and an unwavering commitment to ethical principles.

