# OpenReview forum: "Mol-Instructions: A Large-Scale Biomolecular Instruction Dataset for Large Language Models"
_ICLR.cc/2024/Conference — ICLR 2024 poster_

### Official Review · Reviewer_RHmE · 2023-10-24

**Soundness:** 3 good
**Presentation:** 3 good
**Contribution:** 3 good
**Rating:** 6
**Confidence:** 3

**Summary:**

In this study, the authors present Mol-Instructions, a comprehensive molecular instruction dataset designed specifically for Large Language Models (LLMs). Mol-Instructions incorporates pre-existing tasks, information gleaned from established data sources, and template-based instructions. Rigorous quality control measures have been implemented to ensure the dataset's validity and encompassing diversity.

This dataset encompasses a wide spectrum of tasks, including property prediction, molecular description, protein design, and Q&A, among others. Furthermore, it introduces an LLaMA model finetuned on the collected instructions, showcasing its exceptional performance improvements over the original model across multiple case studies.

**Strengths:**

[Comprehensive Dataset Development] It creates an instruction-tuning dataset that can be used for the community of AI4Science and Large Language Models. This dataset addresses a critical need in the field and serves as a valuable resource for the broader scientific community.

[Performance Improvement with tuned LLaMA Model]: The paper's demonstration of the LoRA-tuned LLaMA model's superior performance over the original model in multiple case studies is a significant strength.

[Broad Range of Applications]: The paper's emphasis is on covering a wide range of applications, including property prediction, molecular description, protein design, etc, which has great potential to be adopted for many downstream applications.

**Weaknesses:**

[Limited Quality Control] In the initial phase of the study, there is an absence of information regarding quality control procedures for the generated task descriptions. A more detailed explanation or discussion of quality control measures, even if briefly mentioned in section 3.2, would enhance the transparency and reliability of the research.

[Less Meaningful Experiments] While the performance improvements observed when fine-tuning the LLaMa model on the collected data are promising, it's essential to acknowledge that this setup resembles a supervised learning scenario. Consequently, the comprehensiveness of the performance assessment may be somewhat limited. To provide a more comprehensive evaluation, additional comparisons with domain-specific Large Language Models (LLMs), such as those mentioned in references [1] and [2], would be valuable. Additionally, incorporating more challenging datasets such as ScienceQA [3] could further strengthen the paper's empirical results.

[Insufficient Analysis on Text Generation] The paper would benefit from a more in-depth analysis of the factuality of the generated text.

[1] https://huggingface.co/stanford-crfm/BioMedLM

[2] https://huggingface.co/chaoyi-wu/PMC_LLAMA_7B

[3] https://proceedings.neurips.cc/paper_files/paper/2022/file/11332b6b6cf4485b84afadb1352d3a9a-Paper-Conference.pdf

**Questions:**

1. Could the authors provide more details about the quality control measures applied to the generated task descriptions in their study?

2. Are there plans to include comparisons with domain-specific Large Language Models (LLMs) to provide a more robust evaluation?

3. Is it possible to incorporate more challenging datasets from the ScienceQA to improve the paper's findings and conclusions?

---

> ### Author Response · Authors · 2023-11-16
>
> We sincerely thank you for your insightful feedback. We have addressed your concerns below and hope our responses provide clarity:
>
> **1. Quality control**
>
> In light of your valuable feedback, we have now expanded Section 3.2 to include a more comprehensive explanation of the quality control procedures we implemented **(Highlighted in Section 3.2, Page 4)**.
>
> **2. More domain-specific baselines**
>
> In response, we have incorporated two new domain-specific Large Language Models (LLMs) into our evaluation: Galactica-6.7B[1] and PMC_LLaMA_13B[2]. This is because BioMedLM[3] and PMC_LLAMA_7B[4] are primarily suited for text-generation tasks. When prompted with questions and options, they tend to generate more options rather than providing answers, due to their lack of instruction tuning. To address this, we included PMC_LLaMA_13B, an instruction-tuned version of PMC_LLAMA_7B, as a new baseline in our study.
>
> The updated results reflecting these expansions can be found in **Tables 3 and 4**, as well as **Figures 5, 6, 8, 9, 10, and 11** in our revised manuscript.
>
> **3. More challenging datasets**
>
> Regarding the inclusion of the ScienceQA[5] dataset, we note that the majority of ScienceQA's biology-related content is multimodal, which does not align well with the current focus of Mol-instructions on text and molecular/protein sequences.
>
> Incorporating multimodal data from ScienceQA would require a substantial extension of the Mol-instructions dataset to support formats beyond text and sequences. While this falls outside the scope of our current research, we recognize the potential value of such an expansion. It represents a promising direction for future work, where Mol-instructions could be enhanced to support a broader range of data types, including multimodal data. This would allow for a more comprehensive evaluation of LLMs in biomolecular research and is a pathway we are keen to explore in our future endeavors **(Highlighted in Section 6, Page 9)**.
>
> **4. Factuality analysis**
>
> Thank you for highlighting the importance of a more in-depth analysis of the factuality of the generated text. The main challenge we encounter is the lack of a unified and widely accepted standard for evaluating tasks related to molecular/protein understanding. Consequently, we have relied on measurement methods used in previous NLP research.
>
> We also attempted to use GPT-4 with certain predefined rules for scoring, but this approach did not yield stable scores. On the other hand, expert review, while potentially more accurate, is extremely time-consuming. Acknowledging these limitations, we are actively exploring more feasible and effective methods to analyze the factuality of the generated text, aiming to improve the robustness and reliability of our evaluation process.
>
> Thank you once more for your invaluable suggestions, which have significantly helped in enhancing the rigor of our experiments and the completeness of our paper!
>
> [1] https://huggingface.co/facebook/galactica-6.7b
> [2] https://huggingface.co/axiong/PMC_LLaMA_13B
> [3] https://huggingface.co/stanford-crfm/BioMedLM
> [4] https://huggingface.co/chaoyi-wu/PMC_LLAMA_7B
> [5] Learn to Explain: Multimodal Reasoning via Thought Chains for Science Question Answering. 2022.

---

> > ### Comment · Reviewer_RHmE · 2023-11-22
> >
> > Thanks for your response! I don't have other questions at this point.

---

> > > ### Author Response · Authors · 2023-11-22
> > > **Thanks!**
> > >
> > > Thank you immensely for your feedback! We are gratified to know that we have successfully addressed your concerns.

---

### Official Review · Reviewer_SfxB · 2023-10-31

**Soundness:** 3 good
**Presentation:** 3 good
**Contribution:** 4 excellent
**Rating:** 8
**Confidence:** 4

**Summary:**

This paper proposes Mol-Instructions, a new instruction dataset tailored specifically for the biomolecular domain. Mol-Instructions comprises three pivotal segments: molecule-centric instructions, protein-focused instructions, and overarching biomolecular text directives. Each segment is constructed carefully to bolster the interpretative and predictive prowess of LLMs pertaining to intricate biomolecular characteristics and dynamics. Through extensive instruction tuning experiments on LLMs, the authors demonstrate the effectiveness of Mol-Instructions in enhancing large models’ performance in the intricate realm of biomolecular studies, thus fostering progress in the biomolecular research community.

**Strengths:**

1. This paper presents "Mol-Instructions" which is a helpful instruction dataset for LLMs in AI4Science.

2. Via comprehensive evaluations using LLMs, the authors demonstrate the effectiveness of Mol-Instructions in enhancing large models’ performance in the intricate realm of biomolecular research.

3. The paper underscores the importance of reproducibility, guaranteeing open access to both the dataset and the accompanying code. Such transparency empowers the community to verify and expand upon the findings, catalyzing cooperative efforts and propelling new works in biomolecular research.

**Weaknesses:**

The evaluation maybe a small issue, however, I know it is not easy to evaluate the molecule generation tasks.

**Questions:**

See Weakness

---

> ### Author Response · Authors · 2023-11-16
>
> Thank you very much for your valuable suggestion. Below are our responses:
>
> **Evaluation methods**
>
> Indeed, we have made several attempts and efforts in this regard. For instance, we customized rules to enable GPT-4 to score molecules. However, the outcomes were not as satisfactory as we hoped. GPT-4 consistently produced unstable scores, failing to accurately assess the quality of the molecules. Alternative methods like wet-lab experiments or expert reviews, while potentially more accurate, are prohibitively expensive and time-consuming. We are actively seeking more feasible and efficient solutions to address this evaluation challenge.
>
> Thank you again for your insightful feedback!

---

### Official Review · Reviewer_KuQ5 · 2023-10-31

**Soundness:** 4 excellent
**Presentation:** 3 good
**Contribution:** 4 excellent
**Rating:** 8
**Confidence:** 3

**Summary:**

The paper introduces "Mol-Instructions," a novel dataset specifically designed for biomolecular research, with the primary objective of enhancing the effectiveness of Large Language Models (LLMs) within this domain. The dataset is generously shared as an open-source resource, encouraging extensive utilization in future endeavors. It encompasses diverse tasks and incorporates information from existing data sources, complemented by template-driven instructions. Covering a wide range of topics, the dataset encompasses molecular description, property prediction, protein design, question answering, and more. Stringent quality assurance measures have been implemented to ensure the dataset's authenticity and comprehensive diversity.

**Strengths:**

+ The paper introduces "Mol-Instructions," a novel dataset specifically designed for biomolecular research, with the aim of enhancing the capabilities of Large Language Models (LLMs) in the field of AI for Science.

+ The authors employ this dataset to fine-tune an LLM, followed by a comprehensive evaluation of the obtained results.

+ This paper exhibits a well-structured and well-motivated approach. The figures presented are clear and easily comprehensible, facilitating easy understanding and follow-through.

**Weaknesses:**

I do not see a major weakness in this paper. Please see my question below.

**Questions:**

I appreciate the authors' efforts in collecting such a comprehensive benchmark for biomolecular research. However, recent work[1] showed that a large number of instructions may not be necessary to obtain a well-finetuned model.

Therefore, is it possible to investigate how the performance of Large Language Models (LLMs) using Mol-Instructions is affected by varying the number of training data instances, checking the redundancy, and possibly employing different sampling methods?

[1]: LIMA: Less Is More for Alignment

---

> ### Author Response · Authors · 2023-11-16
>
> Thank you very much for your insightful comments. Below are our responses:
>
> **Impact of Reduced Data on LLM Performance**
>
> The main premise of LIMA[1] is that almost all knowledge is acquired during the pre-training phase, and only a limited amount of instruction tuning data is needed to guide LLMs in generating high-quality outputs. For the domain-specific data encompassed in Mol-instructions, LLMs may not have encountered such data during the pre-training stage. Therefore, using fewer instruction data might not adequately equip the model to grasp the response format of this particular dataset. Our goal in constructing Mol-instructions was to comprehensively cover various biomolecular tasks, thereby infusing the model with relevant knowledge. This ensures that even users focusing on specific tasks can effectively utilize this dataset for subsequent research, benefiting the entire community.
> Indeed, the instruction tuning process may not necessarily require the full set of instruction data. I believe your suggestion provides an excellent starting point for future work, which could focus on discussing the most effective selection methods for sampling the most beneficial alignment data.
>
> Thank you again for your constructive suggestions! We hope our responses address your concerns.
>
> [1] LIMA: Less Is More for Alignment. 2023.

---

### Official Review · Reviewer_aY9N · 2023-11-07

**Soundness:** 2 fair
**Presentation:** 3 good
**Contribution:** 2 fair
**Rating:** 6
**Confidence:** 4

**Summary:**

This paper introduces a valuable new dataset called Mol-Instructions for instruction-tuning large language models (LLMs) on biomolecular tasks. The authors gather and collect high-quality biomolecular data from various licensed sources, such as PubChem and USPTO. Mol-Instructions covers a diverse range of instruction types across molecules, proteins, and biomedical text, aiming to enhance LLMs' capabilities in the biomolecular domain. Through fine-tuning on the Mol-Instructions dataset, the authors show that LLMs' abilities on diverse biomolecular tasks can be improved.

**Strengths:**

- The proposed dataset, Mol-Instructions, is large-scale and comprehensive, covering a diverse range of tasks across molecules, proteins, and text. I appreciate the significant efforts in curating such a valuable dataset for pushing the research of LLMs for scientific domains.
- Visualization and analysis of key dataset statistics are available, providing insights for other researchers in the field. The long-tail distribution analysis is informative.

* The paper is well-written and provides a comprehensive description of the dataset construction process and baseline experiments.

**Weaknesses:**

**Main concern on experiments:**

My major concern is the experiment part. Specifically, the current comparison and presentation scheme can lead to overestimated results and misleading conclusions because it ignores some important domain-adapted baselines and mainly compares them with domain-agnostic baselines, which are not fine-tuned or designed for scientific tasks. The authors are encouraged to appropriately acknowledge and compare these tasks with previous methods and datasets in order to improve their potential impact. Here I list the important baselines below:

1. An easier molecule instruction tuning dataset: Text+Chem T5 [1]. [1] has included most of the tasks/datasets in Mol-Instructions, including forward reaction prediction, retrosynthesis, description-guided molecule design, and molecule description generation. However, the discussion and comparison to Text+Chem T5 are missing.
2. Scientific language models: Galactica [2]. Compared to the general-purpose LLMs (e.g., Vicuna and Alpaca) reported in the experiments, Galactica is specifically designed and pretrained for the science corpus and should be considered an important baseline.
3. Molecular Language Models: MoMu [3] and MolT5 [4]. These models are designed for description-guided molecule design and molecule description generation
4. The recent methods for reagent prediction [7], forward reaction prediction [5,6], and retrosynthesis [5]. Note that, the baselines in these domains employ different evaluation metrics compared to those reported in the paper.
5. Baselines for protein-related tasks.

I understand that incorporating all these methods for empirical comparison is challenging, and an LLM is not expected to outperform all the state-of-the-art methods in every task to be a valuable contribution. However, I think the comparison to Text+Chem T5 [1] and Galactica [2] is essential and important, considering their similar purposes to this submission. It would also be valuable to report the performances of the single-task state-of-the-art methods in the Tables and Figures (you may not have to outperform these methods) to help empirical comparisons in follow-up works.


**Other weaknesses:**

- The quality control and curation process for constructing the dataset should be described more thoroughly. More details in data processing would strengthen confidence in the dataset.
- Table 5 lists the sources of data used in this paper, but it does not detail the availability or the methods for programmatic access to these data sources. Disclosing these particulars may aid researchers in replicating this paper.
- The Mol-Instructions dataset integrates various tasks in the field of biochemistry, such as the molecular and protein-related characteristics shown in Table 2. However, the experiments only conduct tests on a limited number of biomolecular tasks. Such an evaluation approach may not fully reflect performance on complex biomolecular problems.



Reference:

[1] Unifying Molecular and Textual Representations via Multi-task Language Modelling. In ICML 2023.

[2] Galactica: A Large Language Model for Science. 2022

[3] A Molecular Multimodal Foundation Model Associating Molecule Graphs with Natural Language. 2023

[4] Translation between Molecules and Natural Language. In EMNLP 2022.

[5] Root-aligned SMILES: a tight representation for chemical reaction prediction. In Chemical Science 2022.

[6] Chemformer: a pre-trained transformer for computational chemistry. In *Mach. Learn.: Sci. Technol.* 2022.

[7] Reagent prediction with a molecular transformer improves reaction data quality. In chemical science.

**Questions:**

- Have the authors considered any other representation formats besides SELFIES for molecules? I'm curious how will SELFIES perform against alternative descriptors.
- The authors use BLEU, Levenshtein distance, and other text similarity metrics for evaluating molecule generation. Would more domain-specific metrics better depict the molecule generation capacity of the models?
- Do you intend to expand Mol-Instructions with more task types and data in the future? What directions are you considering to continue improving it as a resource?

**Details Of Ethics Concerns:**

No.

---

> ### Author Response · Authors · 2023-11-16
>
> We sincerely thank you for your insightful feedback. Below are our detailed responses to your concerns:
>
> **1. More domain-specific baselines**
>
> Based on your recommendations, we have expanded our set of baselines to include Text+Chem T5[1], Galactica[2], and MolT5[3] as these models have undergone instruction tuning, aligning them more closely with the objectives of our study.
>
> Regarding the baselines [4,5,6], we have chosen not to include them in our comparison as they focus on pretraining purely on molecular data without any instruction-following. This difference in approach makes a direct comparison somewhat unfair. However, we would like to clarify that our evaluation methods for reagent prediction, forward reaction prediction, and retrosynthesis are consistent with those used by Text+Chem T5[1], which we hope alleviates some of your concerns.
>
> For protein-related tasks, the most closely related baseline we found was [7]. However, due to the unavailability of their code, we were unable to include it in our direct comparison. Instead, we have incorporated Galactica[2] as a baseline for these tasks, considering its relevance and accessibility.
>
> The updated results reflecting these expansions can be found in **Tables 3 and 4**, as well as **Figures 5, 6, 8, 9, 10, and 11** in our revised manuscript.
>
> **2. Lack of data processing details**
>
> Due to page constraints, a detailed description of our data transformation process, from the original sources to instruction data, was not feasible within the main text. However, to ensure transparency and reproducibility, we have included an extensive explanation in **Appendix B (Pages 15-22)**. This section thoroughly details each task's definition, data sources, and the methodology employed to convert this data into the intended instruction format. We believe this provides a clear and comprehensive understanding of our data processing approach. Moreover, we have also added further details on quality control measures **(Highlighted in Section 3.2, Page 4)**.
>
> **3. Insufficient tasks**
>
> We apologize for any misunderstanding and would like to clarify that our experiments comprehensively cover the 17 tasks listed in Figure 3. These include:
>
> - **Molecule-oriented Tasks:** Molecular Description Generation (Figure 5), Description-guided Molecule Design (Table 4), Forward Reaction Prediction (Table 4), Retrosynthesis (Table 4), Reagent Prediction (Table 4), Property Prediction (Table 3).
> - **Protein-oriented Tasks:** Domain/Motif Prediction (Figure 5), Functional Description Generation (Figure 5), Protein Function Prediction (Figure 5), Catalytic Activity Prediction (Figure 5), Protein Design (Figure 9)
> - **Biomolecular Text Tasks:** Open Question (Figure 6), Multi-choice Question (Figure 6), Chemical-protein Interaction Extraction (Figure 6), Chemical-disease Interaction Extraction (Figure 6), Chemical Entity Recognition (Figure 6), True or False Question (Figure 6).
>
> We believe that the variety of tasks covered by Mol-Instructions provides a substantial representation of common biomolecular challenges. Nevertheless, we recognize that there are aspects of biomolecular research that we have not yet explored. These represent opportunities for further enhancement in future research, and we are committed to continually expanding and refining our dataset to encompass an even wider range of biomolecular challenges.
>
> Thank you again for your valuable suggestions! Your input has significantly contributed to enhancing the fairness of our experimental comparisons and the overall completeness of our paper.
>
> [1] Unifying Molecular and Textual Representations via Multi-task Language Modelling. In ICML 2023.
> [2] Galactica: A Large Language Model for Science. 2022.
> [3] Translation between Molecules and Natural Language. In EMNLP 2022.
> [4] Root-aligned SMILES: a tight representation for chemical reaction prediction. In Chemical Science 2022.
> [5] Chemformer: a pre-trained transformer for computational chemistry. In Mach. Learn.: Sci. Technol. 2022.
> [6] Reagent prediction with a molecular transformer improves reaction data quality. In chemical science.
> [7] ProteinDT: A Text-guided Protein Design Framework. 2023.

---

> ### Comment · Reviewer_aY9N · 2023-11-22
> **One more question**
>
> Thanks for your response. It has solved most of my concerns.
> I have one more question to add. As I read through the revision, in Table 4, new baselines of MolT5 and Text+CHEM T5 are divided from other baselines. Is this presentation appropriate and fair? Considering MolT5 and Text+Chem T5 are also language models, I do not see specific differences to other models.
>
> Authors are encouraged to revise the presentation in Table 4, and I will raise the rating accordingly.

---

> > ### Author Response · Authors · 2023-11-22
> > **Modifications to Table 4**
> >
> > Thank you for your insightful feedback.
> >
> > The initial categorization in Table 4 was intended to underscore MolT5 and Text+Chem T5 as domain-specific smaller models, since these models often outperform large language models due to their specialized nature.
> >
> > Following your suggestion, we have revised Table 4 in our manuscript. We believe this update presents a clearer comparison and hope it more appropriately addresses your concern.
> >
> > Thank you once again for your valuable input in enhancing the quality of our work.

---

> > ### Author Response · Authors · 2023-11-22
> > **Thanks!**
> >
> > Thank you very much for your feedback! We are pleased that we could address your concerns.

---

### Author Response · Authors · 2023-11-16
**Global Response (1/2)**

Dear all reviewers,

We are deeply grateful for your valuable time and insightful feedback. A revised draft of our manuscript has been uploaded, with changes highlighted in magenta font for ease of reference. Below, we summarize the main revisions:

**1. More Domain-Specific Baselines**

To enhance the completeness and comprehensiveness of our experiments, we have added several new baselines: Text+Chem T5 [1], Galactica [2], and MolT5 [3] for molecule-oriented tasks, Galactica [2] for protein-related tasks, and Galactica [2] along with PMC_LLaMA [4] for biomolecular text tasks. The experimental results are conveniently presented in the following tables, with a detailed analysis available in **Section 5.1 (Pages 7-9) and Appendix F (Pages 24-28)**.

***1.1 Molecular Description Generation***
|Metric|BLEU-2↑|BLEU-4↑|ROUGE-1↑|ROUGE-2↑|ROUGE-L↑|METEOR↑|
|-|-|-|-|-|-|-|
|Alpaca|0.068|0.014|0.178|0.041|0.136|0.107|
|Baize|0.064|0.015|0.189|0.053|0.148|0.106|
|ChatGLM|0.055|0.011|0.163|0.036|0.121|0.105|
|LLaMa|0.059|0.014|0.164|0.066|0.148|0.184|
|Vicuna|0.052|0.011|0.151|0.055|0.130|0.168|
|**Galactica**|0.024|0.008|0.074|0.015|0.063|0.065|
|Ours|0.217|0.143|0.337|0.196|0.291|0.254|
|**Text+Chem T5**|0.062|0.036|0.126|0.075|0.119|0.139|
|**MolT5**|0.002|0.001|0.036|0.001|0.034|0.033|

***1.2 Property Prediction***
|Metric|MAE↓|
|-|-|
|Alpaca|322.109|
|Baize|261.343|
|ChatGLM|-|
|LLaMa|5.553|
|Vicuna|860.051|
|**Galactica**|0.568|
|Ours|0.013|

***1.3 Description-guided Molecule Design***
|Metric|Exact↑|BLEU↑|Levenshtein↓|RDK FTS↑|MACC FTS↑|Morgan FTS↑|Validity↑|
|-|-|-|-|-|-|-|-|
|Alpaca|0.000|0.004|51.088|0.006|0.029|0.000|0.002|
|Baize|0.000|0.006|53.796|0.000|0.000|0.000|0.002|
|ChatGLM|0.000|0.004|53.157|0.005|0.000|0.000|0.005|
|LLaMa|0.000|0.003|59.864|0.005|0.000|0.000|0.003|
|Vicuna|0.000|0.006|60.356|0.006|0.001|0.000|0.001|
|**Galactica**|0.000|0.192|44.152|0.135|0.238|0.088|0.992|
|Ours|0.002|0.345|41.367|0.231|0.412|0.147|1.000|
|**Text+Chem T5**|0.097|0.508|41.819|0.352|0.474|0.353|0.721|
|**MolT5**|0.112|0.546|38.276|0.400|0.538|0.295|0.773|

***1.4 Reagent Prediction***
|Metric|Exact↑|BLEU↑|Levenshtein↓|RDK FTS↑|MACC FTS↑|Morgan FTS↑|Validity↑|
|-|-|-|-|-|-|-|-|
|Alpaca|0.000|0.026|29.037|0.029|0.016|0.001|0.186|
|Baize|0.000|0.051|30.628|0.022|0.018|0.004|0.099|
|ChatGLM|0.000|0.019|29.169|0.017|0.006|0.002|0.074|
|LLaMa|0.000|0.003|28.040|0.037|0.001|0.001|0.001|
|Vicuna|0.000|0.010|27.948|0.038|0.002|0.001|0.007|
|**Galactica**|0.000|0.141|30.760|0.036|0.127|0.051|0.995|
|Ours|0.044|0.224|23.167|0.237|0.364|0.213|1.000|
|**Text+Chem T5**|0.000|0.225|49.323|0.039|0.186|0.052|0.313|

***1.5 Forward Reaction Prediction***
|Metric|Exact↑|BLEU↑|Levenshtein↓|RDK FTS↑|MACC FTS↑|Morgan FTS↑|Validity↑|
|-|-|-|-|-|-|-|-|
|Alpaca|0.000|0.065|41.989|0.004|0.024|0.008|0.138|
|Baize|0.000|0.044|41.500|0.004|0.025|0.009|0.097|
|ChatGLM|0.000|0.183|40.008|0.050|0.100|0.044|0.108|
|LLaMa|0.000|0.020|42.002|0.001|0.002|0.001|0.039|
|Vicuna|0.000|0.057|41.690|0.007|0.016|0.006|0.059|
|**Galactica**|0.000|0.468|35.021|0.156|0.257|0.097|0.946|
|Ours|0.045|0.654|27.262|0.313|0.509|0.262|1.000|
|**Text+Chem T5**|0.239|0.782|20.413|0.705|0.789|0.652|0.762|

***1.6 Retrosynthesis***
|Metric|Exact↑|BLEU↑|Levenshtein↓|RDK FTS↑|MACC FTS↑|Morgan FTS↑|Validity↑|
|-|-|-|-|-|-|-|-|
|Alpaca|0.000|0.063|46.915|0.005|0.023|0.007|0.160|
|Baize|0.000|0.095|44.714|0.025|0.050|0.023|0.112|
|ChatGLM|0.000|0.117|48.365|0.056|0.075|0.043|0.046|
|LLama|0.000|0.036|46.844|0.018|0.029|0.017|0.010|
|Vicuna|0.000|0.057|46.877|0.025|0.030|0.021|0.017|
|**Galactica**|0.000|0.452|34.940|0.167|0.274|0.134|0.984|
|Ours|0.009|0.705|31.227|0.283|0.487|0.230|1.000|
|**Text+Chem T5**|0.141|0.765|24.043|0.685|0.765|0.585|0.698|

***1.7 Protein Understanding***
|Task|Protein Function|Functional Description|Catalytic Activity|Domain/Motif|
|-|-|-|-|-|
|**Metric**|**ROUGE-L**↑|**ROUGE-L**↑|**ROUGE-L**↑|**ROUGE-L**↑|
|Alpaca|0.20|0.10|0.23|0.12|
|Baize|0.20|0.15|0.22|0.13|
|ChatGLM|0.15|0.14|0.13|0.10|
|LLaMa|0.12|0.12|0.13|0.09|
|Vicuna|0.15|0.14|0.16|0.12|
|**Galactica**|0.07|0.08|0.08|0.06|
|Ours|0.43|0.44|0.52|0.46|

---

> ### Author Response · Authors · 2023-11-16
> **Global Response (2/2)**
>
> ***1.8 Q&A and Information Extraction***
> |Task|True or False|Multi-choice|Chemical Entity Recognition|Chemical-disease Interaction Extraction|Chemical-protein Interaction Extraction|
> |-|-|-|-|-|-|
> |**Metric**|**Acc**↑|**Acc**↑|**F1**↑|**F1**↑|**F1**↑|
> |Alpaca|0.330|0.286|0.213|0.037|0.002|
> |Baize|0.480|0.237|0.009|0.004|0.004|
> |ChatGLM|0.180|0.223|0.150|0.020|0.003|
> |LLaMa|0.270|0.297|0.000|0.050|0.003|
> |Vicuna|0.120|0.290|0.024|0.084|0.013|
> |**Galactica**|0.420|0.312|0.166|0.026|0.001|
> |**PMC_LLaMa**|0.510|0.625|0.003|0.000|0.000|
> |Ours|0.550|0.649|0.753|0.399|0.224|
>
> ***1.9 Open Question***
> |Metric|BLEU↑|ROUGE-1↑|BertScore↑|
> |-|-|-|-|
> |Alpaca|0.003|0.088|0.824|
> |Baize|0.005|0.100|0.811|
> |ChatGLM|0.003|0.090|0.795|
> |LLaMa|0.003|0.100|0.814|
> |Vicuna|0.004|0.097|0.814|
> |**Galactica**|0.000|0.039|0.794|
> |**PMC_LLaMA**|0.007|0.788|0.625|
> |Ours|0.024|0.221|0.837|
>
> Mol-Instructions enhances the **molecular understanding** capabilities of LLMs, demonstrating notable improvements in every metric compared to baseline models, including even domain-specific smaller models. However, the **molecule generation** capabilities of LLMs still exhibit a discernible gap when compared to specialized smaller models, mainly because LLMs are designed to handle a wider range of tasks at the expense of the specialized performance seen in more focused models. For **protein understanding**, the model fine-tuned with Mol-Instructions outperforms other LLMs. And for **biotext NLP tasks**, it even surpasses PMC_LLaMA, which was specifically tuned with instructions on medical texts.
>
> **2. Data Processing and Quality Control**
>
> Due to page constraints, the detailed description of our instruction data construction process for each task is thoroughly outlined in **Appendix B (Pages 15-22)**. Additionally, we have augmented **Section 2.3** with further details on the quality control of task descriptions.
>
> In conclusion, we extend our heartfelt thanks to all the reviewers for their invaluable suggestions, which have significantly enhanced the rigor and completeness of our work. We welcome further discussion and are more than willing to answer any additional questions you may have. Thank you very much!
>
> Sincerely,
> Authors
>
> [1] Unifying Molecular and Textual Representations via Multi-task Language Modelling. In ICML 2023.
> [2] Galactica: A Large Language Model for Science. 2022.
> [3] Translation between Molecules and Natural Language. In EMNLP 2022.
> [4] PMC-LLaMA: Towards Building Open-source Language Models for Medicine. 2023.

---

### Author Response · Authors · 2023-11-21
**Urgent Request for Re-review and Discussion**

Dear Reviewers and AC,

We genuinely value the constructive comments and insightful suggestions you provided for our work. Recognizing the approaching end of the discussion period on **November 22nd**, we kindly urge you to participate in the ongoing discussion and provide any additional insights or clarifications you may have. Your expertise is invaluable to us, and we believe your input will significantly contribute to the improvement of our work.

Thank you very much for your time and consideration. We look forward to hearing from you soon.

Authors

---

### Meta-Review · Program_Chairs · 2023-12-06

**Metareview:**

The paper introduces Mol-Instructions, a dataset designed to enhance Large Language Models' (LLMs) proficiency in the specialized domain of biomolecular studies. The dataset will largely benefit the biomolecular research community.

All the reviewers are positive about the paper.

**Justification For Why Not Higher Score:**

While this paper is commendable, it primarily centers on the creation of a dataset and offers limited innovation or contribution in terms of methods or theory.

**Justification For Why Not Lower Score:**

The dataset will largely benefit the biomolecular research community

---

### Decision · Program_Chairs · 2024-01-16

Accept (poster)